# Modeling Latent Neural Dynamics with Gaussian Process Switching Linear Dynamical Systems

**Amber Hu**
Stanford University
amberhu@stanford.edu

**David Zoltowski**
Stanford University
dzoltow@stanford.edu

**Aditya Nair**
Caltech & Howard Hughes Medical Institute
adi.nair@caltech.edu

**David Anderson**
Caltech & Howard Hughes Medical Institute
wuwei@caltech.edu

**Lea Duncker**\*
Columbia University
ld3149@columbia.edu

**Scott Linderman**\*
Stanford University
swl1@stanford.edu

## Abstract

Understanding how the collective activity of neural populations relates to computation and ultimately behavior is a key goal in neuroscience. To this end, statistical methods which describe high-dimensional neural time series in terms of low-dimensional latent dynamics have played a fundamental role in characterizing neural systems. Yet, what constitutes a successful method involves two opposing criteria: (1) methods should be expressive enough to capture complex nonlinear dynamics, and (2) they should maintain a notion of interpretability often only warranted by simpler linear models. In this paper, we develop an approach that balances these two objectives: the *Gaussian Process Switching Linear Dynamical System* (gpSLDS). Our method builds on previous work modeling the latent state evolution via a stochastic differential equation whose nonlinear dynamics are described by a Gaussian process (GP-SDEs). We propose a novel kernel function which enforces smoothly interpolated locally linear dynamics, and therefore expresses flexible – yet interpretable – dynamics akin to those of recurrent switching linear dynamical systems (rSLDS). Our approach resolves key limitations of the rSLDS such as artifactual oscillations in dynamics near discrete state boundaries, while also providing posterior uncertainty estimates of the dynamics. To fit our models, we leverage a modified learning objective which improves the estimation accuracy of kernel hyperparameters compared to previous GP-SDE fitting approaches. We apply our method to synthetic data and data recorded in two neuroscience experiments and demonstrate favorable performance in comparison to the rSLDS.

## 1 Introduction

Computations in the brain are thought to be implemented through the dynamical evolution of neural activity. Such computations are typically studied in a controlled experimental setup, where an animal is engaged in a behavioral task with relatively few relevant variables. Consistent with this, empirical neural activity has been reported to exhibit many fewer degrees of freedom than there are neurons in the measured sample during such simple tasks [1]. These observations have driven the use of latent

---

\*Corresponding authors.

38th Conference on Neural Information Processing Systems (NeurIPS 2024).

variable models to characterize low-dimensional structure in high-dimensional neural population activity [2, 3]. In this setting, neural activity is often modeled in terms of a low-dimensional latent state that evolves with Markovian dynamics [4–14]. It is thought that the latent state evolution is related to the computation of the system, and therefore, insights into how this evolution is shaped through a dynamical system can help us understand the mechanisms underlying computation [15–21].

In practice, choosing an appropriate modeling approach for a given task requires balancing two key criteria. First, statistical models should be expressive enough to capture potentially complex and nonlinear dynamics required to carry out a particular computation. On the other hand, these models should also be interpretable and allow for straightforward post-hoc analyses of dynamics. One model class that strikes this balance is the recurrent switching linear dynamical system (rSLDS) [8]. The rSLDS approximates arbitrary nonlinear dynamics by switching between a finite number of linear dynamical systems. This leads to a powerful and expressive model which maintains the interpretability of linear systems. Because of their flexibility and interpretability, variants of the rSLDS have been used in many neuroscience applications [19, 22–28] and are examples of a general set of models aiming to understand nonlinear dynamics using compact and interpretable components [29, 30].

However, rSLDS models suffer from several limitations. First, while the rSLDS is a probabilistic model, typical use cases do not capture posterior uncertainty over inferred dynamics. This makes it difficult to judge the extent to which particular features of a fitted model should be relied upon when making inferences about their role in neural computation. Second, the rSLDS often suffers from producing oscillatory dynamics in regions of high uncertainty in the latent space, such as boundaries between linear dynamical regimes. This artifactual behavior can significantly impact the interpretability and predictive performance of the rSLDS. Lastly, the rSLDS does not impose smoothness or continuity assumptions on the dynamics due to its discrete switching formulation. Such assumptions are often natural and useful in the context of modeling realistic neural systems.

In this paper, we improve upon the rSLDS by introducing the *Gaussian Process Switching Linear Dynamical System* (gpSLDS). Our method extends prior work on the Gaussian process stochastic differential equation (GP-SDE) model, a continuous-time method that places a Gaussian process (GP) prior on latent dynamics. By developing a novel GP kernel function, we enforce locally linear, interpretable structure in dynamics akin to that of the rSLDS. Our framework addresses the aforementioned modeling limitations of the rSLDS and contributes a new class of priors in the GP-SDE model class. Our paper is organized as follows. Section 2 provides background on GP-SDE and rSLDS models. Section 3 presents our new gpSLDS model and an inference and learning algorithm for fitting these models. In Section 4 we apply the gpSLDS to a synthetic dataset and two datasets from real neuroscience experiments to demonstrate its practical use and competitive performance. We review related work in Section 5 and conclude our paper with a discussion in Section 6.

## 2 Background

### 2.1 Gaussian process stochastic differential equation models

Gaussian processes (GPs) define nonparametric distributions over functions. They are a popular choice in machine learning due to their ability to capture nonlinearities and encode reasonable prior assumptions such as smoothness and continuity [31]. Here, we review the GP-SDE, a Bayesian generative model that leverages the expressivity of GPs for inferring latent dynamics [10].

**Generative model**   In a GP-SDE, the evolution of the latent state $\boldsymbol{x} \in \mathbb{R}^K$ is modeled as a continuous-time SDE which underlies observed neural activity $\boldsymbol{y}(t_i) \in \mathbb{R}^D$ at time-points $t_i \in [0, T]$. Mathematically, this is expressed as

$$d\boldsymbol{x} = \boldsymbol{f}(\boldsymbol{x})dt + \boldsymbol{\Sigma}^{\frac{1}{2}}d\boldsymbol{w}, \qquad \mathbb{E}[\boldsymbol{y}(t_i) \mid \boldsymbol{x}] = g\left(\boldsymbol{C}\boldsymbol{x}(t_i) + \boldsymbol{d}\right). \tag{1}$$

The drift function $\boldsymbol{f} : \mathbb{R}^K \to \mathbb{R}^K$ describes the system dynamics, $\boldsymbol{\Sigma}$ is a noise covariance matrix, and $d\boldsymbol{w} \sim \mathcal{N}(\boldsymbol{0}, dt\boldsymbol{I})$ is a Wiener process increment. Parameters $\boldsymbol{C} \in \mathbb{R}^{D \times K}$ and $\boldsymbol{d} \in \mathbb{R}^D$ define an affine mapping from latent to observed space, which is then passed through a pre-specified inverse link function $g(\cdot)$.

A GP prior is used to model each output dimension of the dynamics $\boldsymbol{f}(\cdot)$ independently. More formally, if $\boldsymbol{f}(\cdot) = [f_1(\cdot), \ldots, f_K(\cdot)]^\mathsf{T}$, then

$$f_k(\cdot) \overset{\text{iid}}{\sim} \mathcal{GP}(0, \kappa^\Theta(\cdot, \cdot)), \quad \text{for } k = 1, \cdots, K, \tag{2}$$

where $\kappa^\Theta(\cdot, \cdot)$ is the kernel for the GP with hyperparameters $\Theta$.

**Interpretability**   GP-SDEs and their variants can infer complex nonlinear dynamics with posterior uncertainty estimates in physical systems across a variety of applications [32–34]. However, one limitation of using this method with standard GP kernels, such as the radial basis function (RBF) kernel, is that its expressivity leads to dynamics that are often challenging to interpret. In Duncker et al. [10], this was addressed by conditioning the GP prior of the dynamics $\boldsymbol{f}(\cdot)$ on fixed points $\boldsymbol{f}(\boldsymbol{x}^*) = \boldsymbol{0}$ and their local Jacobians $J(\boldsymbol{x}^*) = \frac{\partial}{\partial \boldsymbol{x}} \boldsymbol{f}(\boldsymbol{x})|_{\boldsymbol{x}=\boldsymbol{x}^*}$, and subsequently learning the fixed-point locations $\boldsymbol{x}^*$ and the locally-linearized dynamics $J(\boldsymbol{x}^*)$ as model parameters. This approach allows for direct estimation of key features of $\boldsymbol{f}(\cdot)$. However, due to its flexibility, it is also prone to finding more fixed points than those included in the prior conditioning, which undermines its overall interpretability.

## 2.2   Recurrent switching linear dynamical systems

The rSLDS models nonlinear dynamics by switching between different sets of linear dynamics [8]. Accordingly, it retains the simplicity and interpretability of linear dynamical systems while providing much more expressive power. For these reasons, variations of the rSLDS are commonly used to model neural dynamics [19, 22–29].

**Generative model**   The rSLDS is a discrete-time generative model of the following form:

$$\boldsymbol{x}_t \sim \mathcal{N}(\boldsymbol{A}_{s_t} \boldsymbol{x}_{t-1} + \boldsymbol{b}_{s_t}, \boldsymbol{Q}_{s_t}), \qquad \mathbb{E}[\boldsymbol{y}_t \mid \boldsymbol{x}_t] = g(\boldsymbol{C}\boldsymbol{x}_t + \boldsymbol{d}) \tag{3}$$

where dynamics switch between $J$ distinct linear systems with parameters $\{\boldsymbol{A}_j, \boldsymbol{b}_j, \boldsymbol{Q}_j\}_{j=1}^J$. This is controlled by a discrete state variable $s_t \in \{1, \ldots, J\}$, which evolves via transition probabilities modeled by a multiclass logistic regression,

$$p(s_t \mid s_{t-1}, \boldsymbol{x}_{t-1}) \propto \exp(\boldsymbol{w}^\mathsf{T} \boldsymbol{x}_{t-1} + r_{s_{t-1}}). \tag{4}$$

The "recurrent" nature of this model comes from the dependence of eq. (4) on latent space locations $\boldsymbol{x}_t$. As such, the rSLDS can be understood as learning a partition of the latent space into $J$ linear dynamical regimes seprated by linear decision boundaries. This serves as important motivation for the parametrization of the gpSLDS, as we describe later.

**Interpretability**   While the rSLDS has been successfully used in many applications to model nonlinear dynamical systems, it suffers from a few practical limitations. First, it often produces unnatural artifacts of modeling nonlinear dynamics with discrete switches between linear systems. For example, it may oscillate between discrete modes with different discontinuous dynamics when a trajectory is near a regime boundary. Next, common fitting techniques for rSLDS models with non-conjugate observations typically treat dynamics as learnable hyperparameters rather than as probabilistic quantities [26], which prevents the model from being able to capture posterior uncertainty over the learned dynamics. Inferring a posterior distribution over dynamics is especially important in many neuroscience applications, where scientists often draw conclusions from discovering key features in latent dynamics, such as fixed points or line attractors.

## 3   Gaussian process switching linear dynamical systems

To address these limitations of the rSLDS, we propose a new class of models called the *Gaussian Process Switching Linear Dynamical System* (gpSLDS). The gpSLDS combines the modeling advantages of the GP-SDE with the structured flexbility of the rSLDS. We achieve this balance by designing a novel GP kernel function that defines a smooth, locally linear prior on dynamics. While our main focus is on providing an alternative to the rSLDS, the gpSLDS also contributes a new prior which allows for more interpretable learning of dynamics and fixed points than standard priors in the GP-SDE framework (e.g., the RBF kernel). Our implementation of the gpSLDS is available at: https://github.com/lindermanlab/gpslds.

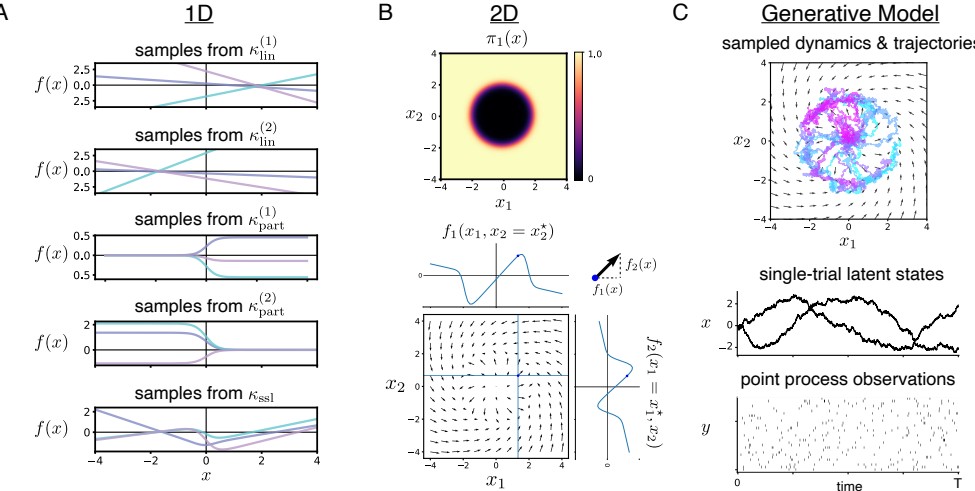

Figure 1: SSL kernel and generative model. **A.** 1D function samples, plotted in different colors, from GPs with five kernels: two linear kernels with different hyperparameters, partition kernels for each of the two regimes, and the SSL kernel. **B.** (*top*) An example $\boldsymbol{\pi}(\boldsymbol{x})$ in 2D and (*bottom*) a sample of dynamics from a SSL kernel in 2D with $\boldsymbol{\pi}(\boldsymbol{x})$ as hyperparameters. The $x_1$- and $x_2$- directions of the arrows are given by independent 1D samples of the kernel. **C.** Schematic of the generative model. Simulated trajectories follow the sampled dynamics. Each trajectory is observed via Poisson process or Gaussian observations.

### 3.1 The smoothly switching linear kernel

The core innovation of our method is a novel GP kernel, which we call the *Smoothly Switching Linear* (SSL) kernel. The SSL kernel specifies a GP prior over dynamics that maintains the switching linear structure of rSLDS models, while allowing dynamics to smoothly interpolate between different linear regimes.

For every pair of locations $\boldsymbol{x}, \boldsymbol{x}' \in \mathbb{R}^K$, the SSL kernel with $J$ linear regimes is defined as,

$$\kappa_{\text{ssl}}(\boldsymbol{x}, \boldsymbol{x}') = \sum_{j=1}^{J} \underbrace{((\boldsymbol{x} - \boldsymbol{c}_j)^\mathsf{T} \boldsymbol{M}(\boldsymbol{x}' - \boldsymbol{c}_j) + \sigma_0^2)}_{\kappa_{\text{lin}}^{(j)}(\boldsymbol{x}, \boldsymbol{x}')} \underbrace{\pi_j(\boldsymbol{x})\pi_j(\boldsymbol{x}')}_{\kappa_{\text{part}}^{(j)}(\boldsymbol{x}, \boldsymbol{x}')} \tag{5}$$

where $\boldsymbol{c}_j \in \mathbb{R}^K$, $\boldsymbol{M} \in \mathbb{R}^{K \times K}$ is a diagonal positive semi-definite matrix, $\sigma_0^2 \in \mathbb{R}_+$, and $\pi_j(\boldsymbol{x}) \geq 0$ with $\sum_{j=1}^{J} \pi_j(\boldsymbol{x}) = 1$. To gain an intuitive understanding of the SSL kernel, we will separately analyze each of the two terms in the summands.

The first term, $\kappa_{\text{lin}}^{(j)}(\boldsymbol{x}, \boldsymbol{x}')$, is a standard linear kernel which defines a GP distribution over linear functions [31]. The superscript $j$ denotes that it is the linear prior on the dynamics in regime $j$. $\boldsymbol{M}$ controls the variance of the function's slope in each input dimension, and $\boldsymbol{c}_j$ is such that the variance of the function achieves its minimum value of $\sigma_0^2$ at $\boldsymbol{x} = \boldsymbol{c}_j$. We expand on the relationship between the linear kernel and linear dynamical systems in Appendix A.

The second term is what we define as the *partition kernel*, $\kappa_{\text{part}}^{(j)}(\boldsymbol{x}, \boldsymbol{x}')$, which gives the gpSLDS its switching structure. We interpret $\boldsymbol{\pi}(\boldsymbol{x}) = [\pi_1(\boldsymbol{x}) \quad \ldots \quad \pi_J(\boldsymbol{x})]^\mathsf{T}$ as parametrizing a categorical distribution over $J$ linear regimes akin to the discrete switching variables in the rSLDS. We model $\boldsymbol{\pi}(\boldsymbol{x})$ as a multiclass logistic regression with decision boundaries $\{\boldsymbol{w}_j^\mathsf{T} \boldsymbol{\phi}(\boldsymbol{x}) = 0 \mid j = 1, \ldots, J-1\}$, where $\boldsymbol{\phi}(\boldsymbol{x})$ is any feature transformation of $\boldsymbol{x}$. This yields random functions which are locally constant and smoothly interpolate at the decision boundaries. More formally,

$$\pi_j(\boldsymbol{x}) = \frac{\exp(\boldsymbol{w}_j^\mathsf{T} \boldsymbol{\phi}(\boldsymbol{x})/\tau)}{1 + \sum_{j=1}^{J-1} \exp(\boldsymbol{w}_j^\mathsf{T} \boldsymbol{\phi}(\boldsymbol{x})/\tau)}, \quad j = 1, \ldots, J \tag{6}$$

where $\boldsymbol{w}_J = 0$. The hyperparameter $\tau \in \mathbb{R}_+$ controls the smoothness of the decision boundaries. As $\tau \to 0$, $\boldsymbol{\pi}(\boldsymbol{x})$ approaches a one-hot vector which produces piecewise constant functions with

sharp boundaries, and as $\tau \to \infty$ the boundaries become more uniform. While we focus on the parametrization in eq. (6) for the experiments in this paper, we note that in principle any classification method can be used, such as another GP or a neural network.

The SSL kernel in eq. (5) naturally combines aspects of the linear and partition kernels via sums and products of kernels, which has an intuitive interpretation [35]. The product kernel $\kappa_{\text{lin}}^{(j)}(\boldsymbol{x}, \boldsymbol{x}')\kappa_{\text{part}}^{(j)}(\boldsymbol{x}, \boldsymbol{x}')$ enforces linearity in regions where $\pi_j(\boldsymbol{x})$ is close to 1. Summing over $J$ regimes then enforces linearity in each of the $J$ regimes, leading to a prior on locally linear functions with knots determined by $\boldsymbol{\pi}(\boldsymbol{x})$. We note that our kernel is reminiscent of the one in Pfingsten et al. [36], which uses a GP classifier as a prior for $\boldsymbol{\pi}(\boldsymbol{x})$ and applies their kernel to a GP regression setting. Here, our work differs in that we explicitly enforce linearity in each regime and draw a novel connection to switching models like the rSLDS.

Figure 1A depicts 1D samples from each kernel. Figure 1B shows how a SSL kernel with $J = 2$ linear regimes separated by decision boundary $x_1^2 + x_2^2 = 4$ *(top)* produces a structured 2D flow field consisting of two linear systems, with $x_1$- and $x_2$- directions determined by 1D function samples *(bottom)*.

## 3.2 The gpSLDS generative model

The full generative model for the gpSLDS incorporates the SSL kernel in eq. (5) into a GP-SDE modeling framework. Instead of placing a GP prior with a standard kernel on the system dynamics as in eq. (2), we simply plug in our new SSL kernel so that

$$f_k(\cdot) \overset{\text{iid}}{\sim} \mathcal{GP}(0, \kappa_{\text{ssl}}^{\Theta}(\cdot, \cdot)), \quad \text{for } k = 1, \cdots, K, \tag{7}$$

where the kernel hyperparameters are $\Theta = \{\boldsymbol{M}, \sigma_0^2, \{\boldsymbol{c}_j\}_{j=1}^J, \{\boldsymbol{w}_j\}_{j=1}^{J-1}, \tau\}$. We then sample latent states and observations according to the GP-SDE via eq. (1). A schematic of the full generative model is depicted in fig. 1C.

**Incorporating inputs** In many modern neuroscience applications, we are often interested in how external inputs to the system, such as experimental stimuli, influence latent states. To this end, we also consider an extension of the model in eq. (1) which incorporates additive inputs of the form,

$$d\boldsymbol{x} = (\boldsymbol{f}(\boldsymbol{x}) + \boldsymbol{B}\boldsymbol{v}(t))dt + \boldsymbol{\Sigma}^{\frac{1}{2}}d\boldsymbol{w}, \tag{8}$$

where $\boldsymbol{v}(t) \in \mathbb{R}^I$ is a time-varying, known input signal and $\boldsymbol{B} \in \mathbb{R}^{K \times I}$ maps inputs linearly to the latent space. The latent path inference and learning approaches presented in the following section can naturally be extended to this setting, with updates for $\boldsymbol{B}$ available in closed form. Further details are provided in Appendix B.3-B.6.

## 3.3 Latent path inference and parameter learning

For inference and learning in the gpSLDS, we apply and extend a variational expectation-maximization (vEM) algorithm for GP-SDEs from Duncker et al. [10]. In particular, we propose a modification of this algorithm that dramatically improves the learning accuracy of kernel hyperparameters, which are crucial to the interpretability of the gpSLDS. We outline the main ideas of the algorithm here, though full details can be found in Appendix B.

As in Duncker et al. [10], we consider a factorized variational approximation to the posterior,

$$q(\boldsymbol{x}, \boldsymbol{f}, \boldsymbol{u}) = q(\boldsymbol{x}) \prod_{k=1}^{K} p(f_k \mid \boldsymbol{u}_k, \Theta)q(\boldsymbol{u}_k), \tag{9}$$

where we have augmented the model with sparse inducing points to make inference of $\boldsymbol{f}$ tractable [37]. The inducing points are located at $\{\boldsymbol{z}_m\}_{m=1}^{M} \subset \mathbb{R}^K$ and take values $(f_k(\boldsymbol{z}_1), \ldots, f_k(\boldsymbol{z}_M))^{\top} = \boldsymbol{u}_k$. Standard vEM maximizes the evidence lower bound (ELBO) to the marginal log-likelihood $\log p(\boldsymbol{y} \mid \Theta)$ by alternating between updating the variational posterior $q$ and updating model hyperparameters $\Theta$ [38]. Using the factorization in eq. (9), we will denote the ELBO as $\mathcal{L}(q(\boldsymbol{x}), q(\boldsymbol{u}), \Theta)$.

For inference of $q(\boldsymbol{x})$, we follow the approach first proposed by Archambeau et al. [39] and extended by Duncker et al. [10]. Computing the ELBO using this approach requires computing variational

expectations of the SSL kernel, which we approximate using Gauss-Hermite quadrature as they are not available in closed form. Full derivations of this step are provided in Appendix B.3. For inference of $q(\boldsymbol{u})$, we follow Duncker et al. [10] and choose a Gaussian variational posterior for $q(\boldsymbol{u}_k) = \mathcal{N}(\boldsymbol{u}_k \mid \boldsymbol{m}_u^{k*}, \boldsymbol{S}_u^{k*})$, which admits closed-form updates for the mean $\boldsymbol{m}_u^{k*}$ and covariance $\boldsymbol{S}_u^{k*}$ given $q(\boldsymbol{x})$ and $\Theta$. Duncker et al. [10] perform these updates before updating $\Theta$ via gradient ascent on the ELBO in each vEM iteration.

In our setting, this did not work well in practice. The gpSLDS often exhibits strong dependencies between $q(\boldsymbol{u})$ and $\Theta$, which makes standard coordinate-ascent steps in vEM prone to getting stuck in local maxima. These dependencies arise due to the highly structured nature of our GP prior; small changes in the decision boundaries $\{\boldsymbol{w}_j\}_{j=1}^{J-1}$ can lead to large (adverse) changes in the prior, which prevents vEM from escaping suboptimal regions of parameter space. To overcome these difficulties, we propose a different approach for learning $\Theta$: instead of fixing $q(\boldsymbol{u})$ and performing gradient ascent on the ELBO, we optimize $\Theta$ by maximizing a partially optimized ELBO,

$$\Theta^* = \arg\max_{\Theta} \left\{ \max_{q(\boldsymbol{u})} \mathcal{L}(q(\boldsymbol{x}), q(\boldsymbol{u}), \Theta) \right\}. \tag{10}$$

Due to the conjugacy of the model, the inner maximization can be performed analytically. This approach circumvents local optima that plague coordinate ascent on the standard ELBO. While other similar approaches exploit model conjugacy for faster vEM convergence in sparse variational GPs [37, 40] and GP-SDEs [41], our approach is the first to our knowledge that leverages this structure specifically for learning the latent dynamics of a GP-SDE model. We empirically demonstrate the superior performance of our learning algorithm in Appendix C.

### 3.4 Recovering predicted dynamics

It is straightforward to obtain the approximate posterior distribution over $\boldsymbol{f}^* := \boldsymbol{f}(\boldsymbol{x}^*)$ evaluated at any new location $\boldsymbol{x}^*$. Under the assumption that $\boldsymbol{f}^*$ only depends on the inducing points, we can use the approximation $q(\boldsymbol{f}^*) = \prod_{k=1}^{K} \int p(f_k^* \mid \boldsymbol{u}_k, \Theta) q(\boldsymbol{u}_k) \, \mathrm{d}\boldsymbol{u}_k$ which can be computed in closed-form using properties of conditional Gaussian distributions. For a batch of points $\{\boldsymbol{x}_i^*\}_{i=1}^N$, this can be computed in $O(NM^2)$ time. The full derivation can be found in Appendix B.4.

This property highlights an appealing feature of the gpSLDS over the rSLDS. The gpSLDS infers a posterior distribution over dynamics at every point in latent space, even in regions of high uncertainty. Meanwhile, as we shall see later, the rSLDS expresses uncertainty by randomly oscillating between different sets of most-likely linear dynamics, which is much harder to interpret.

## 4 Results

### 4.1 Synthetic data

We begin by applying the gpSLDS to a synthetic dataset consisting of two linear rotational systems, one clockwise and one-counterclockwise, which combine smoothly at $x_1 = 0$ (fig. 2A). We simulate 30 trials of latent states from an SDE as in eq. (1) and then generate Poisson process observations given these latent states for $D = 50$ output dimensions (i.e. neurons) over $T = 2.5$ seconds (Fig. 2B). To initialize $\boldsymbol{C}$ and $\boldsymbol{d}$, we fit a Poisson LDS [4] to data binned at 20ms with identity dynamics. For the rSLDS, we also bin the data at 20ms. We then fit the gpSLDS and rSLDS models with $J = 2$ linear states using 5 different random initializations for 100 vEM iterations, and choose the fits with the highest ELBOs in each model class.

We find that the gpSLDS accurately recovers the true latent trajectories (fig. 2C) as well as the true rotation dynamics and the decision boundary between them (fig. 2D). We determine this decision boundary by thresholding the learned $\boldsymbol{\pi}(\boldsymbol{x})$ at 0.5. In addition, we can obtain estimates of fixed point locations by computing the posterior probability $\prod_{k=1}^{K} \mathbb{P}_{q(\boldsymbol{f})}(|f_k(\boldsymbol{x})| < \epsilon)$ for a small $\epsilon > 0$; the locations $\boldsymbol{x}$ with high probability are shaded in purple. This reveals that the gpSLDS finds high-probability fixed points which overlap significantly with the true fixed points, denoted by stars. In comparison, both the rSLDS and the GP-SDE with RBF kernel do not learn the correct decision boundary nor the fixed points as accurately (fig. 2E-F). Of particular note, the RBF kernel incorrectly extrapolates and finds a superfluous fixed point outside the region traversed by the true latent states.

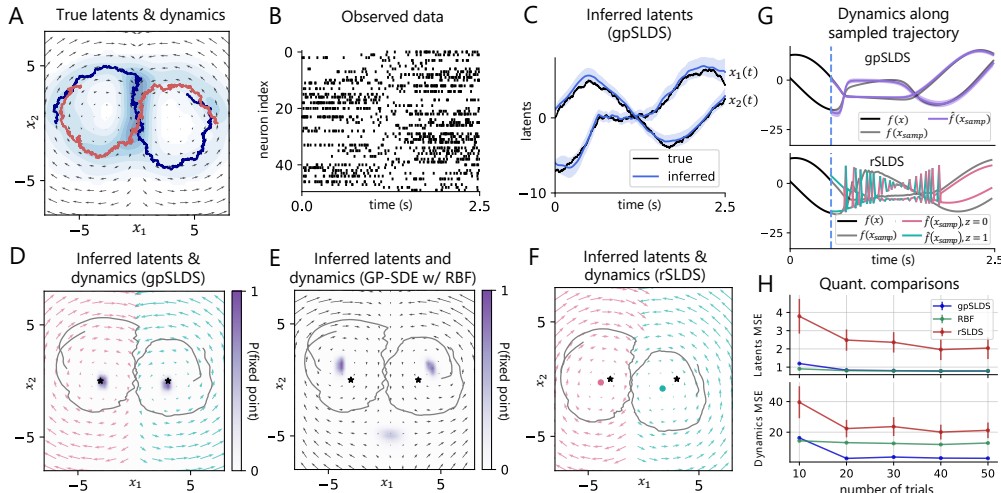

Figure 2: Synthetic data results. **A.** True dynamics and latent states used to generate the dataset. Dynamics are clockwise and counterclockwise linear systems separated by $x_1 = 0$. Two latent trajectories are shown on top of a kernel density estimate of the latent states visited by all 30 trials. **B.** Poisson process observations from an example trial. **C.** True vs. inferred latent states for the gpSLDS and rSLDS, with 95% posterior credible intervals. **D.** Inferred dynamics (pink/green) and two inferred latent trajectories (gray) corresponding to those in Panel A from a gpSLDS fit with 2 linear regimes. The model finds high-probability fixed points (purple) overlapping with true fixed points (stars). **E.** Analogous plot to D for the GP-SDE model with RBF kernel. Note that this model does not provide a partition of the dynamics. **F.** rSLDS inferred latents, dynamics, and fixed points (pink/green dots). **G.** *(top)* Sampled latents and corresponding dynamics from the gpSLDS, with 95% posterior credible intervals. *(bottom)* Same, but for the rSLDS. The pink/green trace represents the most likely dynamics at the sampled latents, colored by discrete switching variable. *H.* MSE between true and inferred latents and dynamics for gpSLDS, GP-SDE with RBF kernel, and rSLDS while varying the number of trials in the dataset. Error bars are $\pm 2$SE over 5 random initializations.

Figure 2G illustrates the differences in how the gpSLDS and the rSLDS express uncertainty over dynamics. To the left of the dashed line, we sample latent states starting from $x_0 = (7, 0)$ and plot the corresponding true dynamics. To the right, we simulate latent states $x_{\mathsf{samp}}$ from the fitted model and plot the true dynamics (in gray) and the learned most likely dynamics (in color) at $x_{\mathsf{samp}}$. For a well-fitting model, we would expect the true and learned dynamics to overlap. We see that the gpSLDS produces smooth simulated dynamics that match the true dynamics at $x_{\mathsf{samp}}$ *(top)*. By contrast, the rSLDS expresses uncertainty by oscillating between the two linear dynamical systems, hence producing uninterpretable dynamics at $x_{\mathsf{samp}}$ *(bottom)*. This region of uncertainty overlaps with the $x_1 = 0$ boundary, suggesting that the rSLDS fails to capture the smoothly interpolating dynamics present in the true system.

Next, we perform quantitative comparisons between the three competing methods (fig. 2H). We find that both continuous-time methods consistently outperform the rSLDS on both metrics, suggesting that these methods are likely more suitable for modeling Poisson process data. Moreover, the gpSLDS better recovers dynamics compared to the RBF kernel, illustrating that the correct inductive bias can yield performance gains over a more flexible prior, especially in a data-limited setting.

Lastly, we note that the gpSLDS can achieve more expressive power than the rSLDS by learning *nonlinear* decision boundaries between linear regimes, for instance by incorporating nonlinear features into $\phi(x)$ in eq. (6). We demonstrate this feature for a 2D limit cycle in Appendix D.

### 4.2 Application to hypothalamic neural population recordings during aggression

In this section, we revisit the analyses of Nair et al. [27], which applied dynamical systems models to neural recordings during aggressive behavior in mice. To do this, we reanalyze a dataset which consists of calcium imaging of ventromedial hypothalamus neurons from a mouse interacting with two consecutive intruders. The recording was collected from 104 neurons at 15 Hz over $\sim$343

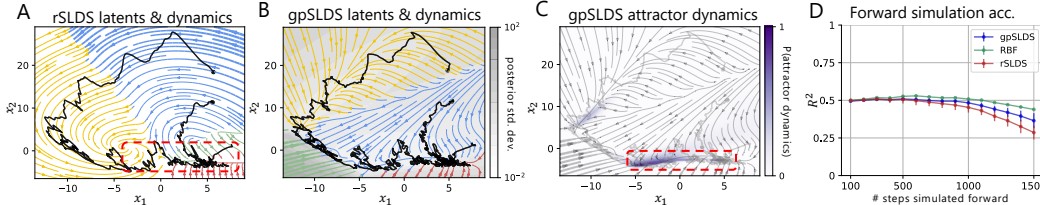

Figure 3: Results on hypothalamic data from Nair et al. [27]. In each of the panels A-C, flow field arrow widths are scaled by the magnitude of dynamics for clarity of visualization. **A.** rSLDS inferred latents and most likely dynamics. The presumed location of the line attractor from [27] is marked with a red box. **B.** gpSLDS inferred latents and most likely dynamics in latent space. Background is colored by posterior standard deviation of dynamics averaged across latent dimensions, which adjusts relative to the presence of data in the latent space. **C.** Posterior probability of slow points in gpSLDS, which validates line-attractor like dynamics, as marked by a red box. **D.** Comparison of in-sample forward simulation $R^2$ between gpSLDS, rSLDS, and GP-SDE with RBF kernel. To compute this, we choose initial conditions uniformly spaced 100 time bins apart in both trials, simulate latent states $k$ steps forward according to learned dynamics (with $k$ ranging from 100-1500), and evaluate the $R^2$ between predicted and true observations as in Nassar et al. [23]. Error bars are $\pm 2$SE over 5 different initializations.

seconds (i.e. 5140 time bins). Nair et al. [27] found that an rSLDS fit to this data learns dynamics that form an approximate line attractor corresponding to an aggressive internal state (fig. 3A). Here, we supplement this analysis by using the gpSLDS to directly assess model confidence about this finding.

For our experiments, we $z$-score and then split the data into two trials, one for each distinct intruder interacting with the mouse. Following Nair et al. [27], we choose $J = 4$ linear regimes to compare the gpSLDS and rSLDS. We choose $K = 2$ latent dimensions to aid the visualization of the resulting model fits; we find that even in such low dimensions, both models still recover line attractor-like dynamics. We model the calcium imaging traces as Gaussian emissions on an evenly spaced time grid and initialize $C$ and $d$ using principal component analysis. We fit models with 5 different initializations for 50 vEM iterations and display the runs with highest forward simulation accuracy (as described in the caption of fig. 3).

In fig. 3A-B, we find that both methods infer similar latent trajectories and find plausible flow fields that are parsed in terms of simpler linear components. We further demonstrate the ability of the gpSLDS to more precisely identify the line attractor from Nair et al. [27]. To do this, we use the learned $q(\boldsymbol{f})$ to compute the posterior probability of slow dynamics on a dense $(80 \times 80)$ grid of points in the latent space using the procedure in Section 3.4. The gpSLDS finds a high-probability region of slow points corresponding to the approximate line attractor found in Nair et al. [27] (fig. 3C). This demonstrates a key advantage of the gpSLDS over the rSLDS: by modeling dynamics probabilistically using a structured GP prior, we can validate the finding of a line attractor with further statistical rigor. Finally, we compare the gpSLDS, rSLDS, and GP-SDE with RBF kernel using an in-sample forward simulation metric (fig. 3D). All three methods retain similar predictive power 500 steps into the future. After that, the gpSLDS performs slightly worse than the RBF kernel; however, it gains interpretability by imposing piecewise linear structure while still outperforming the rSLDS.

### 4.3 Application to lateral intraparietal neural recodings during decision making

In neuroscience, there is considerable interest in understanding how neural dynamics during decision making tasks support the process of making a choice [26, 34, 42–47]. In this section, we use the gpSLDS to infer latent dynamics from spiking activity in the lateral intraparietal (LIP) area of monkeys reporting decisions about the direction of motion of a random moving dots stimulus with varying degrees of motion coherence [42, 43]. The animal indicated its choice of net motion direction (either left or right) via a saccade. On some trials, a 100ms pulse of weak motion, randomly oriented to the left or right, was also presented to the animal. Here, we model the dynamics of 58 neurons recorded across 50 trials consisting of net motion coherence strengths in $\{-.512, -.128, 0.0, .128, .512\}$, where the sign corresponds to the net movement direction of the stimulus. We only consider data 200ms after motion onset, corresponding to the start of decision-related activity.

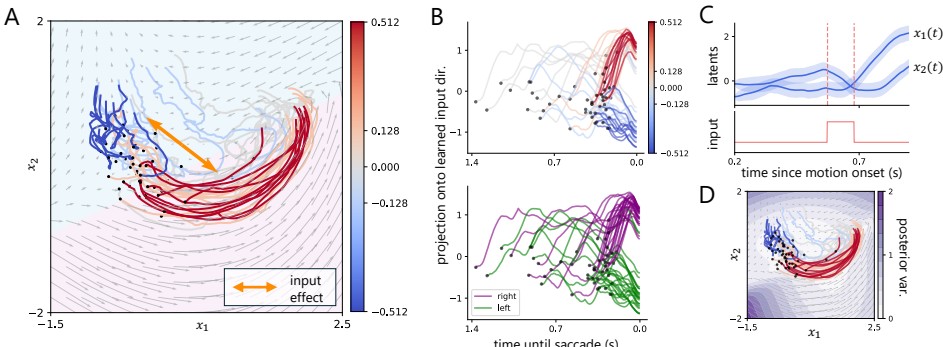

Figure 4: Results on LIP spiking data from a decision-making task in Stine et al. [42]. **A.** gpSLDS inferred latents colored by coherence, inferred dynamics with background colored by most likely linear regime, and the learned input-driven direction depicted by an orange arrow. **B.** Projection of latents onto the 1D input-driven axis from Panel A, colored by coherence *(top)* and choice *(bottom)*. **C.** Inferred latents with 95% credible intervals and corresponding 100ms pulse input for an example trial. **D.** Posterior variance of dynamics produced by the gpSLDS.

To capture potential input-driven effects, we fit a version of the gpSLDS described in eq. (8) with $K = 2$ latent dimensions and $J = 2$ linear regimes over 50 vEM iterations. We encoded the input signal as $\pm 1$ with sign corresponding to pulse direction. In Figure 4A, we find that not only does the gpSLDS capture a distinct visual separation between trials by motion coherence, but it also learns a precise separating decision boundary between the two linear regimes. Our finding is consistent with previous work on a related task, which found that average LIP responses can be represented by a 2D curved manifold [48], though here we take a dynamical systems perspective. Additionally, our model learns an input-driven effect which appears to define a separating axis. To verify this, we project the inferred latent states onto the 1D subspace spanned by the input effect vector. Figure 4B shows that the latent states separate by coherence *(top)* and by choice *(bottom)*, further suggesting that the pulse input relates to meaningful variation in evidence accumulation for this task. Fig. 4C shows an example latent trajectory aligned with pulse input; during the pulse there is a noticeable change in the latent trajectory. Lastly, in Fig. 4D we find that the gpSLDS expresses high confidence in learned dynamics where latent trajectories are present and low confidence in areas further from this region.

## 5 Related work

There are several related approaches to learning nonlinear latent dynamics in discrete or continuous time. Gaussian process state-space models (GP-SSMs) [49–53] can be considered a discrete-time analogue to GP-SDEs. In a GP-SSM, observations are assumed to be regularly sampled and latent states evolve according to a discretized dynamical system with a GP prior. Wang et al. [49] and Turner et al. [50] learned the dynamics of a GP-SSM using maximum a posteriori estimation. Frigola et al. [51] and Eleftheriadis et al. [52] employed a variational approximation with sparse inducing points to infer the latent states and dynamics in a fully Bayesian fashion. In our work, we use a continuous-time framework that more naturally handles irregularly sampled data, such as point-process observations commonly encountered in neural spiking data. Neural ODEs and SDEs [54, 55] use deep neural networks to parametrize the dynamics of a continuous-time system, and have emerged as prominent tools for analyzing large datasets, including those in neuroscience [33, 56–58]. While these methods can represent flexible function classes, they are likely to overfit to low-data regimes and may be difficult to interpret. In addition, unlike the gpSLDS, neural ODEs and SDEs do not typically quantify uncertainty of the learned dynamics.

In the context of dynamical mixture models, Köhs et al. [59, 60] proposed a continuous-time switching model in a GP-SDE framework. This model assumes a latent Markov jump process over time which controls the system dynamics by switching between different SDEs. The switching process models dependence on time, but not location in latent space. In contrast, the gpSLDS does not explicitly represent a latent switching process and rather models switching probabilities as part of the kernel

function. The dependence of the kernel on the location in latent space allows for the gpSLDS to partition the latent space into different linear regimes.

While our work has followed the inference approach of Archambeau et al. [61] and Duncker et al. [10], other methods for latent path inference in nonlinear SDEs have been proposed [32, 41, 62, 63]. Verma et al. [41] parameterized the posterior SDE path using an exponential family-based description. The resulting inference algorithm showed improved convergence of the E-step compared to Archambeau et al. [39]. Course and Nair [32, 62] proposed an amortization strategy that allows the variational update of the latent state to be parallelized over sequence length. In principle, any of these approaches could be applied to inference in the gpSLDS and would be an interesting direction for future work.

## 6   Discussion

In this paper, we introduced the gpSLDS to infer low-dimensional latent dynamical systems from high-dimensional, noisy observations. By developing a novel kernel for GP-SDE models that defines distributions over smooth locally-linear functions, we were able to relate GP-SDEs to rSLDS models and address key limitations of the rSLDS. Using both simulated and real neural datasets, we demonstrated that the gpSLDS can accurately infer true generative parameters and performs favorably in comparison to rSLDS models and GP-SDEs with other kernel choices. Moreover, our real data examples illustrate the variety of potential practical uses of this method. On calcium imaging traces recorded during aggressive behavior, the gpSLDS reveals dynamics consistent with the hypothesis of a line attractor put forward based on previous rSLDS analyses [27]. On a decision making dataset, the gpSLDS finds latent trajectories and dynamics that clearly separate by motion coherence and choice, providing a dynamical systems view consistent with prior studies [42, 48].

In our experiments, we demonstrated the ability of the gpSLDS to recover ground-truth dynamical systems and key dynamical features using fixed settings of hyperparameters: the latent dimensionality $K$ and the number of regimes $J$. For simulated data, we set hyperparameters to their true values; for real data, we chose hyperparameters based on prior studies and did not further optimize these values. However, for most real neural datasets, we do not know the true underlying dimensionality or optimal number of regimes. To tune these hyperparameters, we can resort to standard techniques for model comparison in the neural latent variable modeling literature. Two common evaluation metrics are forward simulation accuracy [23, 27] and co-smoothing performance [6, 64, 65].

While these results are promising, we acknowledge a few limitations of the gpSLDS. First, the memory cost scales exponentially with the size of the latent dimension due to using quadrature methods to approximate expectations of the SSL kernel, which are not available in closed form. This computational limitation renders it difficult to fit the gpSLDS with many (i.e. greater than 3) latent dimensions. One potential direction for future work would be to instead use Monte Carlo methods to approximate kernel expectations for models with larger latent dimensionality. In addition, while both the gpSLDS and rSLDS require choosing a discretization timestep for solving dynamical systems, in practice we find that the gpSLDS requires smaller steps for stable model inference. This allows the gpSLDS to more accurately approximate dynamics with continuous-time likelihoods, at the cost of allocating more time bins during inference. Finally, we acknowledge that traditional variational inference approaches – such as those employed by the gpSLDS– tend to underestimate posterior variance due to the KL-divergence-based objective [38]. Carefully assessing biases introduced by our variational approximation to the posterior would be an important topic for future work.

Overall, the gpSLDS provides a general modeling approach for discovering latent dynamics of noisy measurements in an intepretable and fully probabilistic manner. We expect that our model will be a useful addition to the rSLDS and related methods on future analyses of neural data.

## Acknowledgments and Disclosure of Funding

This work was supported by grants from the NIH BRAIN Initiative (U19NS113201, R01NS131987, & RF1MH133778) and the NSF/NIH CRCNS Program (R01NS130789), as well as fellowships from the Simons Collaboration on the Global Brain, the Wu Tsai Neurosciences Institute, the Alfred P. Sloan Foundation, and the McKnight Foundation. We thank Barbara Engelhardt, Julia Palacios, and the members of the Linderman Lab for helpful feedback throughout this project. The authors have no competing interests to declare.

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

# Supplementary Material

## Table of Contents

## A    Relationship between linear kernel and linear dynamical systems

Here, we draw a mathematical connection between functions sampled from a GP with a linear kernel,

$$f(\cdot) \sim \mathcal{GP}(0, \kappa_{\text{lin}}(\cdot, \cdot)), \qquad \kappa_{\text{lin}}(\boldsymbol{x}, \boldsymbol{x}') = (\boldsymbol{x} - \boldsymbol{c})^{\mathsf{T}} \boldsymbol{M} (\boldsymbol{x} - \boldsymbol{c}) + \sigma_0^2 \tag{11}$$

and functions of the form

$$f(\boldsymbol{x}) = \boldsymbol{\beta}^{\mathsf{T}} \boldsymbol{x} + \beta_0. \tag{12}$$

This connection is useful for understanding the relationship between our model formulation and the typical linear dynamical systems formulation as in the rSLDS. In particular, we will show that the linear kernel in eq. (11) equivalently places a prior on $\boldsymbol{\beta}$ and $\beta_0$ in eq. (12), in a similar manner to Bayesian linear regression.

By definition of a GP, for any finite set of input locations $\{\boldsymbol{x}_i\}_{i=1}^N$, we have

$$\begin{bmatrix} f(\boldsymbol{x}_1) & f(\boldsymbol{x}_2) & \dots & f(\boldsymbol{x}_N) \end{bmatrix}^{\mathsf{T}} \sim \mathcal{N}(\boldsymbol{0}, \Phi) \tag{13}$$

where $\Phi_{ij} = \kappa_{\text{lin}}(\boldsymbol{x}_i, \boldsymbol{x}_j)$. Equivalently, for every pair $i, j = 1, \dots, N$,

$$\text{Cov}(f(\boldsymbol{x}_i), f(\boldsymbol{x}_j)) = (\boldsymbol{x}_i - \boldsymbol{c})^{\mathsf{T}} \boldsymbol{M} (\boldsymbol{x}_j - \boldsymbol{c}) + \sigma_0^2 \tag{14}$$

$$= \boldsymbol{x}_i^{\mathsf{T}} \boldsymbol{M} \boldsymbol{x}_j - \boldsymbol{c}^{\mathsf{T}} \boldsymbol{M} (\boldsymbol{x}_i + \boldsymbol{x}_j) + \boldsymbol{c}^{\mathsf{T}} \boldsymbol{c} + \sigma_0^2 \tag{15}$$

Under eq. (12), treating $\boldsymbol{\beta}$ and $\beta_0$ as random, this would become

$$\text{Cov}(\boldsymbol{\beta}^{\mathsf{T}} \boldsymbol{x}_i + \beta_0, \boldsymbol{\beta}^{\mathsf{T}} \boldsymbol{x}_j + \beta_0) = \boldsymbol{x}_i^{\mathsf{T}} \text{Cov}(\boldsymbol{\beta}, \boldsymbol{\beta}) \boldsymbol{x}_j + \text{Cov}(\beta_0, \boldsymbol{\beta})(\boldsymbol{x}_i + \boldsymbol{x}_j) + \text{Var}(\beta_0) \tag{16}$$

Equation (15) and eq. (16) are equivalent and eq. (13) is satisfied if and only if

$$\begin{bmatrix} \boldsymbol{\beta} \\ \beta_0 \end{bmatrix} \sim \mathcal{N} \left( \boldsymbol{0}, \begin{bmatrix} \boldsymbol{M} & -\boldsymbol{M} \boldsymbol{c} \\ -\boldsymbol{c}^{\mathsf{T}} \boldsymbol{M} & \boldsymbol{c}^{\mathsf{T}} \boldsymbol{c} + \sigma_0^2 \end{bmatrix} \right) \tag{17}$$

This shows that a GP with a linear kernel is equivalent to a Bayesian linear regression model with a prior on coefficients of the form in eq. (17).

## B    Inference and learning

We present the full inference and learning details for the gpSLDS with additive inputs,

$$d\boldsymbol{x} = (\boldsymbol{f}(\boldsymbol{x}) + \boldsymbol{B}\boldsymbol{v}(t))dt + \Sigma^{\frac{1}{2}} d\boldsymbol{w}, \qquad \mathbb{E}[\boldsymbol{y}(t_i) \mid \boldsymbol{x}] = g\left(\boldsymbol{C}\boldsymbol{x}(t_i) + \boldsymbol{d}\right). \tag{18}$$

Our approach primarily follows that of Duncker et al. [10]. We use the following notation from Duncker et al. [10] throughout this section:

- $\langle h(\cdot) \rangle_{p(\cdot)}$ denotes the expectation of $h(\cdot)$ with respect to the distribution $p(\cdot)$.

- $\boldsymbol{K}_{xz} \in \mathbb{R}^{N_1 \times N_2}$ denotes the covariance matrix defined by our kernel $\kappa_{\text{ssl}}(\cdot, \cdot)$ for two batches of points $\{\boldsymbol{x}_i\}_{i=1}^{N_1}$ and $\{\boldsymbol{z}_i\}_{i=1}^{N_2}$. Specifically, $[\boldsymbol{K}_{xz}]_{ij} = \kappa_{\text{ssl}}(\boldsymbol{x}_i, \boldsymbol{z}_j)$. If one of these batches has only one point, e.g. $N_1 = 1$, we denote the corresponding covariance vector as $\boldsymbol{k}_{xz} \in \mathbb{R}^{1 \times N_2}$. Note, these quantities depend on $\Theta$ through the kernel; we omit writing this dependence for brevity.

### B.1    Augmenting the generative model

Following Duncker et al. [10], we first augment our generative model with sparse inducing points $\{\boldsymbol{u}_k\}_{k=1}^K \subset \mathbb{R}^M$ at input locations $\{\boldsymbol{z}_m\}_{m=1}^M \subset \mathbb{R}^D$. Inducing points can be seen as pseudo-observations of $\boldsymbol{f}(\cdot)$ at input locations $\{\boldsymbol{z}_m\}_{m=1}^M$. They allow for tractable inference of $\boldsymbol{f}$ at any new batch of $N$ input locations by reducing computational complexity from $O(N^3)$ to $O(NM^2)$, where typically $M$ is chosen to be much smaller than $N$ [37]. After this augmentation, the joint likelihood of our model becomes

$$p(\boldsymbol{y}, \boldsymbol{x}, \boldsymbol{f}, \boldsymbol{u} \mid \Theta) = p(\boldsymbol{y} \mid \boldsymbol{x}) p(\boldsymbol{x} \mid \boldsymbol{f}) \prod_{k=1}^K p(f_k \mid \boldsymbol{u}_k, \Theta) p(\boldsymbol{u}_k \mid \Theta). \tag{19}$$

Treating $\boldsymbol{u}_k$ as pseudo-observations, we assume the following augmented prior:

$$p(\boldsymbol{u}_k \mid \Theta) = \mathcal{N}(\boldsymbol{u}_k \mid \mathbf{0}, \boldsymbol{K}_{zz}). \tag{20}$$

Then we can view $f_k(\cdot) \mid \boldsymbol{u}_k$ as a new GP conditioned on $\boldsymbol{u}_k$,

$$f_k(\cdot) \mid \boldsymbol{u}_k \sim \mathcal{GP}(\mu_{f_k \mid \boldsymbol{u}_k}(\cdot), \kappa_{f_k \mid \boldsymbol{u}_k}(\cdot, \cdot)), \tag{21}$$

where

$$\mu_{f_k \mid \boldsymbol{u}_k}(\boldsymbol{x}) = \boldsymbol{k}_{xz} \boldsymbol{K}_{zz}^{-1} \boldsymbol{u}_k$$
$$\kappa_{f_k \mid \boldsymbol{u}_k}(\boldsymbol{x}, \boldsymbol{x}') = \kappa_{\mathrm{ssl}}(\boldsymbol{x}, \boldsymbol{x}') - \boldsymbol{k}_{xz} \boldsymbol{K}_{zz}^{-1} \boldsymbol{k}_{zx}.$$

## B.2   Variational lower bound

As in Duncker et al. [10], we consider a variational approximation to the posterior of the form

$$q(\boldsymbol{x}, \boldsymbol{f}, \boldsymbol{u}) = q(\boldsymbol{x}) \prod_{k=1}^{K} p(f_k \mid \boldsymbol{u}_k, \Theta) q(\boldsymbol{u}_k). \tag{22}$$

Using this factorization, we derive the ELBO to the marginal log-likelihood of our model. By Jensen's inequality,

$$
\begin{aligned}
\log p(\boldsymbol{y} \mid \Theta) &= \log \int p(\boldsymbol{y} \mid \boldsymbol{x}) p(\boldsymbol{x} \mid \boldsymbol{f}) p(\boldsymbol{f} \mid \boldsymbol{u}, \Theta) p(\boldsymbol{u} \mid \Theta) d\boldsymbol{x} d\boldsymbol{f} d\boldsymbol{u} \\
&\geq \int q(\boldsymbol{x}, \boldsymbol{f}, \boldsymbol{u}) \log \frac{p(\boldsymbol{y} \mid \boldsymbol{x}) p(\boldsymbol{x} \mid \boldsymbol{f}) p(\boldsymbol{f} \mid \boldsymbol{u}, \Theta) p(\boldsymbol{u} \mid \Theta)}{q(\boldsymbol{x}, \boldsymbol{f}, \boldsymbol{u})} d\boldsymbol{x} d\boldsymbol{f} d\boldsymbol{u} \\
&= \int q(\boldsymbol{x}, \boldsymbol{f}, \boldsymbol{u}) \log \frac{p(\boldsymbol{y} \mid \boldsymbol{x}) p(\boldsymbol{x} \mid \boldsymbol{f}) \prod_{k=1}^{K} p(\boldsymbol{u}_k \mid \Theta)}{q(\boldsymbol{x}) \prod_{k=1}^{K} q(\boldsymbol{u}_k)} d\boldsymbol{x} d\boldsymbol{f} d\boldsymbol{u} \\
&= \langle \log p(\boldsymbol{y} \mid \boldsymbol{x}) \rangle_{q(\boldsymbol{x})} - \langle \mathrm{KL}[q(\boldsymbol{x}) \| p(\boldsymbol{x} \mid \boldsymbol{f})] \rangle_{q(\boldsymbol{f})} - \sum_{k=1}^{K} \mathrm{KL}[q(\boldsymbol{u}_k) \| p(\boldsymbol{u}_k \mid \Theta)] \\
&:= \mathcal{L}(q(\boldsymbol{x}), q(\boldsymbol{u}), \Theta),
\end{aligned}
$$

where

$$q(\boldsymbol{f}) = \prod_{k=1}^{K} \int p(f_k \mid \boldsymbol{u}_k, \Theta) q(\boldsymbol{u}_k) d\boldsymbol{u}_k. \tag{23}$$

## B.3   Inference of latent paths

To perform inference over the posterior of latent paths $q(\boldsymbol{x})$, we follow a method first proposed in Archambeau et al. [39] and extended by Duncker et al. [10].

As in Archambeau et al. [39], we choose a posterior distribution $q(\boldsymbol{x})$ characterized by a Markov Gaussian process,

$$q(\boldsymbol{x}) : \{d\boldsymbol{x} = \underbrace{(-\boldsymbol{A}(t)\boldsymbol{x}(t) + \boldsymbol{b}(t))}_{:= \boldsymbol{f}_q(\boldsymbol{x})} dt + \boldsymbol{\Sigma}^{\frac{1}{2}} d\boldsymbol{w}, \quad \boldsymbol{x}_0 \sim \mathcal{N}(\boldsymbol{m}(0), \boldsymbol{S}(0))\}. \tag{24}$$

This distribution satisfies posterior marginals $q(\boldsymbol{x}_t) = \mathcal{N}(\boldsymbol{x}_t \mid \boldsymbol{m}_t, \boldsymbol{S}_t)$ which satisfy the differential equations

$$\frac{d\boldsymbol{m}(t)}{dt} = -\boldsymbol{A}(t)\boldsymbol{m}(t) + \boldsymbol{b}(t) \tag{25}$$

$$\frac{d\boldsymbol{S}(t)}{dt} = -\boldsymbol{A}(t)\boldsymbol{S}(t) - \boldsymbol{S}(t)\boldsymbol{A}(t)^{\mathsf{T}} + \boldsymbol{\Sigma}. \tag{26}$$

Archambeau et al. [39] maximize the ELBO with respect to $q(\boldsymbol{x})$ subject to the constraints in eq. (25) and eq. (26) using the method of Lagrange multipliers. They show that after applying integration by parts, the Lagrangian becomes

$$\widetilde{\mathcal{L}} = \mathcal{L}(q(\boldsymbol{x}), q(\boldsymbol{u}), \Theta) - \mathcal{C}_1 - \mathcal{C}_2 \tag{27}$$

where

$$\mathcal{C}_1 = \int_0^T \left\{ \boldsymbol{\lambda}(t)^\mathsf{T}(\boldsymbol{A}(t)\boldsymbol{m}(t) - \boldsymbol{b}(t)) - \frac{d\boldsymbol{\lambda}(t)}{dt}^\mathsf{T} \boldsymbol{m}(t) \right\} dt + \boldsymbol{\lambda}(T)^\mathsf{T}\boldsymbol{m}(T) - \boldsymbol{\lambda}(0)^\mathsf{T}\boldsymbol{m}(0) \quad (28)$$

$$\mathcal{C}_2 = \int_0^T \mathrm{Tr}\left[ \boldsymbol{\Psi}(t)(\boldsymbol{A}(t)\boldsymbol{S}(t) + \boldsymbol{S}(t)\boldsymbol{A}(t)^\mathsf{T} - \boldsymbol{\Sigma}) - \frac{d\boldsymbol{\Psi}(t)}{dt}\boldsymbol{S}(t) \right] dt \quad (29)$$

$$+ \mathrm{Tr}[\boldsymbol{\Psi}(T)\boldsymbol{S}(T)] - \mathrm{Tr}[\boldsymbol{\Psi}(0)\boldsymbol{S}(0)] \quad (30)$$

As in Archambeau et al. [39], we assume that $\boldsymbol{\lambda}(T) = \boldsymbol{0}$ and $\boldsymbol{\Psi}(T) = \boldsymbol{0}$.

To find the stationary points of the Lagrangian, we first take derivatives of $\widetilde{\mathcal{L}}$ with respect to $\boldsymbol{m}(0), \boldsymbol{S}(0), \boldsymbol{m}(t), \boldsymbol{S}(t), \boldsymbol{A}(t), \boldsymbol{b}(t)$ and set them to 0. The derivatives with respect to $\boldsymbol{m}(0)$ and $\boldsymbol{S}(0)$ lead to the updates

$$\boldsymbol{m}(0) = \boldsymbol{\mu}(0) - \boldsymbol{V}(0)\boldsymbol{\lambda}(0), \qquad \boldsymbol{S}(0) = \left(2\boldsymbol{\Psi}(0) + \boldsymbol{V}(0)^{-1}\right)^{-1} \quad (31)$$

where we assume a prior on initial conditions $p(\boldsymbol{x}_0) = \mathcal{N}(\boldsymbol{x}_0 \mid \boldsymbol{\mu}(0), \boldsymbol{V}(0))$.

The derivatives with respect to $\boldsymbol{m}(t)$ and $\boldsymbol{S}(t)$ lead to the stationary equations

$$\frac{d\boldsymbol{\lambda}(t)}{dt} = \boldsymbol{A}(t)^\mathsf{T}\boldsymbol{\lambda}(t) - \frac{\partial\mathcal{L}}{\partial\boldsymbol{m}(t)} \quad (32)$$

$$\frac{d\boldsymbol{\Psi}(t)}{dt} = \boldsymbol{A}(t)^\mathsf{T}\boldsymbol{\Psi}(t) - \boldsymbol{\Psi}(t)\boldsymbol{A}(t) - \frac{\partial\mathcal{L}}{\partial\boldsymbol{S}(t)} \odot \mathbb{P} \quad (33)$$

with $\mathbb{P}_{ij} = \frac{1}{2}$ for $i \neq j$ and 1 otherwise. The inclusion of $\mathbb{P}$ was proposed by Duncker et al. [10] to adjust for taking derivatives with respect to a symmetric matrix.

To take derivatives with respect to $\boldsymbol{A}(t)$ and $\boldsymbol{b}(t)$, we first extend a result from Appendix A of Archambeau et al. [39] to our affine inputs model. This allows us to rewrite the KL-divergence term between the posterior and prior latent paths in the ELBO as

$$\langle \mathrm{KL}[q(\boldsymbol{x})||p(\boldsymbol{x} \mid \boldsymbol{f})]\rangle_{q(\boldsymbol{f})} = \frac{1}{2}\int_0^T \left\langle (\boldsymbol{f} + \boldsymbol{B}\boldsymbol{v}(t) - \boldsymbol{f}_q)^\mathsf{T}(\boldsymbol{f} + \boldsymbol{B}\boldsymbol{v}(t) - \boldsymbol{f}_q)\right\rangle_{q(\boldsymbol{x}),q(\boldsymbol{f})} dt$$

$$= \frac{1}{2}\int_0^T \left\langle (\boldsymbol{B}\boldsymbol{v}(t) + \Delta\boldsymbol{f})^\mathsf{T}(\boldsymbol{B}\boldsymbol{v}(t) + \Delta\boldsymbol{f})\right\rangle_{q(\boldsymbol{x}),q(\boldsymbol{f})} dt$$

$$= \frac{1}{2}\int_0^T \left\langle (\Delta\boldsymbol{f})^\mathsf{T}(\Delta\boldsymbol{f})\right\rangle_{q(\boldsymbol{x}),q(\boldsymbol{f})} dt \quad (34)$$

$$+ \int_0^T \boldsymbol{v}(t)^\mathsf{T}\boldsymbol{B}^\mathsf{T} \langle\Delta\boldsymbol{f}\rangle_{q(\boldsymbol{x}),q(\boldsymbol{f})} dt + \frac{1}{2}\int_0^T \boldsymbol{v}(t)^\mathsf{T}\boldsymbol{B}^\mathsf{T}\boldsymbol{B}\boldsymbol{v}(t)dt$$

where $\Delta\boldsymbol{f} := \boldsymbol{f} - \boldsymbol{f}_q$. The integrand of the first term in eq. (34) can be expanded as

$$\left\langle (\Delta\boldsymbol{f})^\mathsf{T}(\Delta\boldsymbol{f})\right\rangle_{q(\boldsymbol{x}),q(\boldsymbol{f})} = \left\langle \boldsymbol{f}^\mathsf{T}\boldsymbol{f}\right\rangle_{q(\boldsymbol{x}),q(\boldsymbol{f})} + 2\mathrm{Tr}\left[\boldsymbol{A}(t)^\mathsf{T}\left\langle\frac{\partial\boldsymbol{f}}{\partial\boldsymbol{x}}\right\rangle_{q(\boldsymbol{x}),q(\boldsymbol{f})}\boldsymbol{S}(t)\right] \quad (35)$$

$$+ \mathrm{Tr}\left[\boldsymbol{A}(t)^\mathsf{T}\boldsymbol{A}(t)(\boldsymbol{S}(t) + \boldsymbol{m}(t)\boldsymbol{m}(t)^\mathsf{T}\right] + 2\boldsymbol{m}(t)^\mathsf{T}\boldsymbol{A}(t)^\mathsf{T}\langle\boldsymbol{f}\rangle_{q(\boldsymbol{x}),q(\boldsymbol{f})}$$

$$+ \boldsymbol{b}(t)^\mathsf{T}\boldsymbol{b}(t) - 2\boldsymbol{b}(t)^\mathsf{T}\langle\boldsymbol{f}\rangle_{q(\boldsymbol{x}),q(\boldsymbol{f})} - 2\boldsymbol{b}^\mathsf{T}\boldsymbol{A}(t)\boldsymbol{m}(t)$$

where we have used the identity $\left\langle \langle\boldsymbol{f}(\boldsymbol{x})\rangle_{q(\boldsymbol{f})}(\boldsymbol{x} - \boldsymbol{m})^\mathsf{T}\right\rangle_{q(\boldsymbol{x})} = \left\langle\frac{\partial\boldsymbol{f}}{\partial\boldsymbol{x}}\right\rangle_{q(\boldsymbol{x}),q(\boldsymbol{f})}\boldsymbol{S}$, which can be derived from Stein's lemma. Note that computing eq. (35) relies on three quantities,

$$\langle\boldsymbol{f}\rangle_{q(\boldsymbol{x}),q(\boldsymbol{f})}, \left\langle\boldsymbol{f}^\mathsf{T}\boldsymbol{f}\right\rangle_{q(\boldsymbol{x}),q(\boldsymbol{f})}, \left\langle\frac{\partial\boldsymbol{f}}{\partial\boldsymbol{x}}\right\rangle_{q(\boldsymbol{x}),q(\boldsymbol{f})}$$

which can be written as terms which depend on expectations of the kernel with respect to $q(\boldsymbol{x})$. We derive these as follows:

$$\langle\boldsymbol{f}\rangle_{q(\boldsymbol{x}),q(\boldsymbol{f})} = \left\langle\boldsymbol{k}_{xz}\boldsymbol{K}_{zz}^{-1}\boldsymbol{u}\right\rangle_{q(\boldsymbol{u}),q(\boldsymbol{x})}$$

$$= \langle\boldsymbol{k}_{xz}\rangle_{q(\boldsymbol{x})}\boldsymbol{K}_{zz}^{-1}\boldsymbol{m}_u$$

$$\langle \boldsymbol{f}^\mathsf{T}\boldsymbol{f}\rangle_{q(\boldsymbol{x}),q(\boldsymbol{f})} = \left\langle \sum_{k=1}^{K} \langle f_k(\boldsymbol{x})^2\rangle_{q(f_k)} \right\rangle_{q(\boldsymbol{x})}$$

$$= \left\langle \sum_{k=1}^{K} \mathrm{Var}_{q(f_k)}[f_k(\boldsymbol{x})] + \langle f_k(\boldsymbol{x})\rangle_{q(f_k)}^2 \right\rangle_{q(\boldsymbol{x})}$$

$$= \sum_{k=1}^{K} \underbrace{\langle \mathrm{Var}_{p(f_k|\boldsymbol{u})}[f_k(\boldsymbol{x})]\rangle_{q(\boldsymbol{x}),q(\boldsymbol{u})}}_{\text{Term 1}} + \underbrace{\langle \mathrm{Var}_{q(u)}[\langle f_k(\boldsymbol{x})\rangle_{p(f_k \mid \boldsymbol{u})}]\rangle_{q(\boldsymbol{x})}}_{\text{Term 2}}$$

$$+ \underbrace{\left\langle \langle f_k(\boldsymbol{x})\rangle_{p(f_k \mid \boldsymbol{u}),q(\boldsymbol{u})}^2 \right\rangle_{q(\boldsymbol{x})}}_{\text{Term 3}}$$

$$= \sum_{k=1}^{K} \underbrace{\left\langle \boldsymbol{k}_{xx} - \boldsymbol{k}_{xz}\boldsymbol{K}_{zz}^{-1}\boldsymbol{k}_{zx}\right\rangle_{q(\boldsymbol{x})}}_{\text{Term 1}} + \underbrace{\left\langle \boldsymbol{k}_{xz}\boldsymbol{K}_{zz}^{-1}\boldsymbol{S}_u^k\boldsymbol{K}_{zz}^{-1}\boldsymbol{k}_{zx}\right\rangle_{q(\boldsymbol{x})}}_{\text{Term 2}}$$

$$+ \underbrace{\left\langle \boldsymbol{k}_{xz}\boldsymbol{K}_{zz}^{-1}\boldsymbol{m}_u^k(\boldsymbol{m}_u^k)^T\boldsymbol{K}_{zz}^{-1}\boldsymbol{k}_{zx}\right\rangle_{q(\boldsymbol{x})}}_{\text{Term 3}}$$

$$\left\langle \frac{\partial \boldsymbol{f}}{\partial \boldsymbol{x}}\right\rangle_{q(\boldsymbol{x}),q(\boldsymbol{f})} = \left\langle \boldsymbol{m}_u^\mathsf{T}\boldsymbol{K}_{zz}^{-1}\tfrac{\partial \boldsymbol{k}_{zx}}{d\boldsymbol{x}}\right\rangle_{q(\boldsymbol{x})}$$

$$= \boldsymbol{m}_u^\mathsf{T}\boldsymbol{K}_{zz}^{-1}\left\langle \tfrac{\partial \boldsymbol{k}_{zx}}{d\boldsymbol{x}}\right\rangle_{q(\boldsymbol{x})}$$

The above three function expectations can thus be expressed in terms of the following four kernel expectations with respect to $q(\boldsymbol{x}) = \mathcal{N}(\boldsymbol{x} \mid \boldsymbol{m}, \boldsymbol{S})$:

$$\langle \kappa(\boldsymbol{x},\boldsymbol{x})\rangle_{q(\boldsymbol{x})}, \quad \langle \kappa(\boldsymbol{x},\boldsymbol{z})\rangle_{q(\boldsymbol{x})}, \quad \langle \kappa(\boldsymbol{z}_1,\boldsymbol{x})\kappa(\boldsymbol{x},\boldsymbol{z}_2)\rangle_{q(\boldsymbol{x})}, \quad \left\langle \frac{\partial \kappa(\boldsymbol{z},\boldsymbol{x})}{\partial \boldsymbol{x}}\right\rangle_{q(\boldsymbol{x})}.$$

For our SSL kernel these kernel expectations are not available in closed form, so in practice we approximate them using Gauss-Hermite quadrature.

Next, differentiating $\widetilde{\mathcal{L}}$ with respect to $\boldsymbol{A}(t)$ and $\boldsymbol{b}(t)$ yields the updates

$$\boldsymbol{A}(t) = - \left\langle \frac{\partial \boldsymbol{f}}{\partial \boldsymbol{x}}\right\rangle_{q(\boldsymbol{x}),q(\boldsymbol{f})} + 2\boldsymbol{\Sigma}\boldsymbol{\Psi}(t) \tag{36}$$

$$\boldsymbol{b}(t) = \langle \boldsymbol{f}(\boldsymbol{x})\rangle_{q(\boldsymbol{x}),q(\boldsymbol{f})} + \boldsymbol{A}(t)\boldsymbol{m}(t) + \boldsymbol{B}\boldsymbol{v}(t) - \boldsymbol{\Sigma}\boldsymbol{\lambda}(t) \tag{37}$$

Note that these updates have one key difference from the previously derived updates in Archambeau et al. [39] and Duncker et al. [10]: the input-dependent term $\boldsymbol{B}\boldsymbol{v}(t)$ in eq. (37). Intuitively, this is because the posterior bias term $\boldsymbol{b}(t)$ is fully time-varying, so it captures changes in the latent states due to input-driven effects in the posterior.

In summary, the inference algorithm for updating $q(\boldsymbol{x})$ is as follows. In each iteration of vEM, we repeat the following forward-backward style algorithm:

1. Solve for $\boldsymbol{m}(t), \boldsymbol{S}(t)$ forward in time starting from $\boldsymbol{m}(0), \boldsymbol{S}(0)$ via eq. (25) and eq. (26).
2. Solve for $\boldsymbol{\lambda}(t), \boldsymbol{\Psi}(t)$ backward in time starting from $\boldsymbol{\lambda}(T), \boldsymbol{\Psi}(T) = \boldsymbol{0}$ via eq. (32) and eq. (33).
3. Update $\boldsymbol{A}(t)$ and $\boldsymbol{b}(t)$ via eq. (36) and eq. (37).

After solving these stationary equations, we update $\boldsymbol{m}(0)$ and $\boldsymbol{S}(0)$ via eq. (31).

**Computational details**   Solving for $\boldsymbol{m}(t), \boldsymbol{S}(t), \boldsymbol{\lambda}(t)$, and $\boldsymbol{\Psi}(t)$ requires integrating continuous-time ODEs. In practice we use Euler integration with a small discretization step $\Delta t$ relative to the sampling rate of the data, though in principle any ODE solver can be used. We found that the ELBO usually converges within 20 forward-backward iterations.

The ODEs for solving $\boldsymbol{\lambda}(t)$ and $\boldsymbol{\Psi}(t)$ in eq. (32) and eq. (33) depend on evaluating gradients of the ELBO with respect to $\boldsymbol{m}(t)$ and $\boldsymbol{S}(t)$. We use modern autodifferentiation capabilities in JAX to compute these gradients.

## B.4   Updating dynamics and hyperparameters with a modified learning objective

As we describe in Section 3.3, our gpSLDS inference algorithm uses a modified objective for hyperparameter learning. In this section, we discuss this objective in detail and present closed-form updates for the inducing points given the hyperparameters. Then, using the inducing points, we will derive the posterior distribution over $\boldsymbol{f}(\cdot)$ at any location in the latent space.

After updating the latent paths $q(\boldsymbol{x})$ as described in Appendix B.3, we update hyperparameters $\Theta$ using a partially optimized ELBO. This update can be written as

$$\Theta^* = \arg\max_{\Theta} \left\{ \max_{q(\boldsymbol{u})} \mathcal{L}(q(\boldsymbol{x}), q(\boldsymbol{u}), \Theta) \right\}. \tag{38}$$

Following Duncker et al. [10], we choose the variational posterior

$$q(\boldsymbol{u}_k) = \mathcal{N}(\boldsymbol{u}_k \mid \boldsymbol{m}_u^k, \boldsymbol{S}_u^k). \tag{39}$$

Given $q(\boldsymbol{x})$ and $\Theta$, this leads to the closed-form updates,

$$\boldsymbol{S}_u^{k*} = \boldsymbol{K}_{zz} \left( \boldsymbol{K}_{zz} + \int_0^T \langle \boldsymbol{k}_{zx}\boldsymbol{k}_{xz} \rangle_{q(\boldsymbol{x})} \, dt \right)^{-1} \boldsymbol{K}_{zz} \tag{40}$$

$$\boldsymbol{m}_u^* = \boldsymbol{S}_u^{k*}\boldsymbol{K}_{zz}^{-1} \int_0^T \left( \langle \boldsymbol{k}_{zx} \rangle_{q(\boldsymbol{x})} \left( -\boldsymbol{A}(t)\boldsymbol{m}(t) + \boldsymbol{b}(t) - \boldsymbol{B}\boldsymbol{v}(t) \right)^\mathsf{T} \right.$$
$$\left. - \left\langle \frac{\partial \boldsymbol{k}_{zx}}{\partial \boldsymbol{x}} \right\rangle_{q(\boldsymbol{x})} \boldsymbol{S}(t)\boldsymbol{A}(t)^\mathsf{T} \right) dt \tag{41}$$

In the above equation, $\boldsymbol{m}_u^* \in \mathbb{R}^{M \times K}$ contains $\boldsymbol{m}_u^{k*} \in \mathbb{R}^M$ in each column. The inside maximization of eq. (38) can be computed analytically using these closed-form updates. Note that $\boldsymbol{m}_u^{k*}$ and $\boldsymbol{S}_u^{k*}$ depend on $\Theta$ through the prior kernel covariances $\boldsymbol{K}_{zz}$ and $\boldsymbol{k}_{zx}$. Therefore, eq. (38) can be understood as performing joint optimization of the ELBO with respect to $\Theta$ through $\boldsymbol{m}_u^{k*}$ and $\boldsymbol{S}_u^{k*}$, as well as through the rest of the ELBO. In practice, this allows vEM to circumvent dependencies between $q(\boldsymbol{u})$ and $\Theta$, leading to more accurate estimation of both quantities. For our experiments, we use the Adam optimizer to solve eq. (38).

After obtaining $\Theta^*$ in each vEM iteration, we explicitly update $q(\boldsymbol{u})$ using eq. (41) and eq. (40) for the next iteration.

**Recovering predicted dynamics**   Given (updated) variational parameters $\boldsymbol{m}_u^k$ and $\boldsymbol{S}_u^k$, it is straightforward to compute the posterior distribution of $\boldsymbol{f}^* := \boldsymbol{f}(\boldsymbol{x}^*)$ at any location $\boldsymbol{x}^*$ in the latent space. Recall the variational approximation from eq. (23). If we apply this approximation to $\boldsymbol{f}^*$, we have

$$q(\boldsymbol{f}^*) = \prod_{k=1}^K q(f_k^*) = \prod_{k=1}^K \int p(f_k^* \mid \boldsymbol{u}_k, \Theta)q(\boldsymbol{u}_k)d\boldsymbol{u}_k. \tag{42}$$

To evaluate this analytically, we use properties of conditional Gaussian distributions. First note that by our augmented prior,

$$p(f_k^* \mid \boldsymbol{u}_k, \Theta) = \mathcal{N}(f_k^* \mid \boldsymbol{k}_{x^*z}\boldsymbol{K}_{zz}^{-1}\boldsymbol{u}_k, \kappa_{\mathrm{ssl}}(\boldsymbol{x}^*, \boldsymbol{x}^*) - \boldsymbol{k}_{x^*z}\boldsymbol{K}_{zz}^{-1}\boldsymbol{k}_{zx^*}). \tag{43}$$

Then, by conjugacy of Gaussian distributions,

$$q(f_k^*) = \int \mathcal{N}(f_k^* \mid \boldsymbol{k}_{x^*z}\boldsymbol{K}_{zz}^{-1}\boldsymbol{u}_k, \kappa_{\mathrm{ssl}}(\boldsymbol{x}^*, \boldsymbol{x}^*) - \boldsymbol{k}_{x^*z}\boldsymbol{K}_{zz}^{-1}\boldsymbol{k}_{zx^*})\mathcal{N}(\boldsymbol{u}_k \mid \boldsymbol{m}_u^k, \boldsymbol{S}_u^k)d\boldsymbol{u}_k$$
$$= \mathcal{N}(f_k^* \mid \boldsymbol{k}_{x^*z}\boldsymbol{K}_{zz}^{-1}\boldsymbol{m}_u^k, \kappa_{\mathrm{ssl}}(\boldsymbol{x}^*, \boldsymbol{x}^*) - \boldsymbol{k}_{x^*z}\boldsymbol{K}_{zz}^{-1}\boldsymbol{k}_{zx^*} + \boldsymbol{k}_{x^*z}\boldsymbol{K}_{zz}^{-1}\boldsymbol{S}_u^k\boldsymbol{K}_{zz}^{-1}\boldsymbol{k}_{zx^*}). \tag{44}$$

## B.5    Learning observation model parameters

For the experiments in this paper, we considered two observation models: Gaussian observations and Poisson process observations.

**Gaussian observations**    We consider the observation model

$$p(\boldsymbol{y} \mid \boldsymbol{x}) = \prod_{t_i} \mathcal{N}(\boldsymbol{y}(t_i) \mid \boldsymbol{C}\boldsymbol{x}(t_i) + \boldsymbol{d}, \boldsymbol{R}). \tag{45}$$

where $\boldsymbol{R} \in \mathbb{R}^D$ is a diagonal covariance matrix. The expected log-likelihood is available in closed form and is given by

$$\langle \log p(\boldsymbol{y} \mid \boldsymbol{x}) \rangle_{q(\boldsymbol{x})} = \sum_{t_i} \left( \log \mathcal{N}(\boldsymbol{y}(t_i) \mid \boldsymbol{C}(t_i) + \boldsymbol{d}, \boldsymbol{R}) - \frac{1}{2}\mathrm{Tr}\left[\boldsymbol{S}(t_i)\boldsymbol{C}^{\mathsf{T}}\boldsymbol{R}^{-1}\boldsymbol{C}\right] \right) \tag{46}$$

Closed-form updates for $\boldsymbol{C}, \boldsymbol{d}$ and $\boldsymbol{R}$ are also available:

$$\boldsymbol{C}^* = \left( \sum_{t_i}(\boldsymbol{y}(t_i) - \boldsymbol{d})\boldsymbol{m}(t_i)^{\mathsf{T}} \right) \left( \sum_{t_i}(\boldsymbol{S}(t_i) + \boldsymbol{m}(t_i)\boldsymbol{m}(t_i)^{\mathsf{T}}) \right)^{-1} \tag{47}$$

$$\boldsymbol{d}^* = \frac{1}{n_{t_i}} \sum_{t_i}(\boldsymbol{y}(t_i) - \boldsymbol{C}^*\boldsymbol{m}(t_i)) \tag{48}$$

$$R_d^* = \frac{1}{n_{t_i}} \sum_{t_i}(y_d(t_i)^2 - 2y_d(t_i)\boldsymbol{c}_d^{\mathsf{T}}\boldsymbol{m}(t_i) + (\boldsymbol{c}_d^{\mathsf{T}}\boldsymbol{m}(t_i))^2 + \boldsymbol{c}_d^{\mathsf{T}}\boldsymbol{S}(t_i)\boldsymbol{c}_d) \tag{49}$$

where $n_{t_i}$ is the number of observed time points, $R_d^*$ is the $d$-th entry of $\boldsymbol{R}$, and $\boldsymbol{c}_d$ is the $d$-th row of $\boldsymbol{C}$.

**Poisson process observations**    The second observation model we consider is Poisson process observations of the form

$$p(\{t_i\} \mid \boldsymbol{x}) = \mathcal{PP}(g(\boldsymbol{C}\boldsymbol{x}(t) + \boldsymbol{d})), \tag{50}$$

where either $g(a) = \exp(a)$ (exponential inverse link) or $g(a) = \log(1 + \exp(a))$ (softplus inverse link). For the exponential inverse link, the expected log-likelihood is available in closed form and is given by

$$\langle \log p(\{t_i\} \mid \boldsymbol{x}) \rangle_{q(\boldsymbol{x})} = -\int_0^T \exp\left( \boldsymbol{C}\boldsymbol{m}(t) + \boldsymbol{d} + \frac{1}{2}\mathrm{diag}(\boldsymbol{C}\boldsymbol{S}(t)\boldsymbol{C}^{\mathsf{T}}) \right) dt + \sum_{t_i}(\boldsymbol{C}\boldsymbol{m}(t_i) + \boldsymbol{d}) \tag{51}$$

For the softplus inverse link, the expected log-likelihood is not available in closed-form, but can be approximated by Gauss-Hermite quadrature or a second-order Taylor expansion around $\boldsymbol{m}(t)$.

For both Poisson process models, we update $\boldsymbol{C}$ and $\boldsymbol{d}$ using gradient ascent on the expected log-likelihood with the Adam optimizer.

## B.6    Learning the input effect matrix

Here we derive a closed-form update for $\boldsymbol{B}$, which linearly maps external inputs to the latent space. The only term in the ELBO which depends on $\boldsymbol{B}$ is $\langle \mathrm{KL}[q(\boldsymbol{x})\|p(\boldsymbol{x} \mid \boldsymbol{f})]\rangle_{q(\boldsymbol{f})}$. We differentiate this term as written in eq. (34) and arrive at the update

$$\boldsymbol{B}^* = -\left( \int_0^T (\langle \boldsymbol{f} \rangle_{q(\boldsymbol{x}),q(\boldsymbol{f})} + \boldsymbol{A}(t)\boldsymbol{m}(t) - \boldsymbol{b}(t))\boldsymbol{v}(t)^{\mathsf{T}} dt \right) \left( \int_0^T \boldsymbol{v}(t)\boldsymbol{v}(t)^{\mathsf{T}} dt \right)^{-1}. \tag{52}$$

Note that the term $\left( \int_0^T \boldsymbol{v}(t)\boldsymbol{v}(t)^{\mathsf{T}} dt \right)^{-1}$ can be pre-computed since $\boldsymbol{v}(t)$ is known.

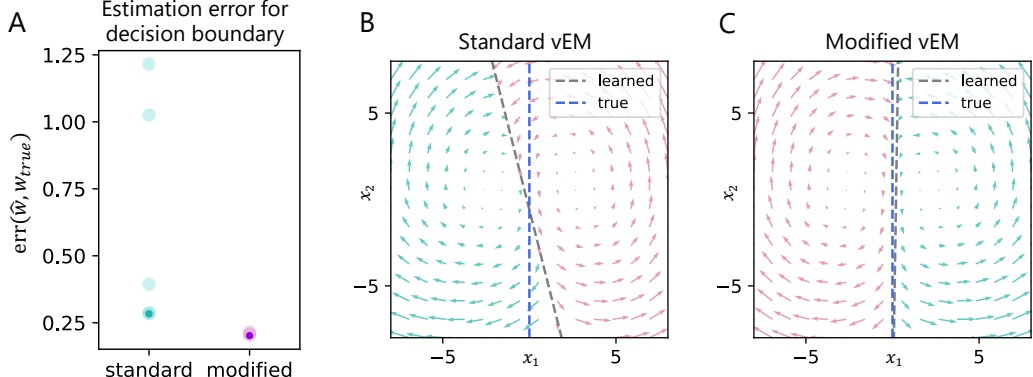

Figure 5: Comparison between the standard vEM approach in Duncker et al. [10] and our modified vEM approach. **A.** Estimation error between the true and learned decision boundaries, computed as described in Appendix C. For each vEM approach, we fit 5 gpSLDS models with different random initializations. The estimation errors are denoted in light blue/purple dots. The runs that we display in the next two panels are denoted by a solid blue/purple dot. **B.** The standard vEM approach fails to learn the true decision boundary of $x_1 = 0$. **C.** The modified vEM approach precisely learns this decision boundary.

## C  Empirical results for new learning objective

In this section, we empirically compare the standard vEM approach from Duncker et al. [10] to our modified vEM approach in which we learn kernel hyperparameters on a partially optimized ELBO. For this experiment, we use the same synthetic dataset from our main result in Section 4.1. We fit the gpSLDS with 5 different random initializations using both standard vEM and modified vEM. For these fits, we fix the values of $C$ and $d$ throughout learning to ensure that the resulting models are anchored to the same latent subspace (in general, they are not guaranteed to end up in the same subspace due to rotational unidentifiability). Each run was fit with 50 total vEM iterations; each iteration consisted of 15 forward-backward solves to update $q(x)$ and 300 Adam gradient steps with a learning rate of 0.01 to update kernel hyperparameters.

To compare the quality of the learned hyperparameters, we quantitatively assess the error between the learned and true decision boundaries. In this simple example with $J = 2$, the decision boundary can be parametrized as $w_0 + w_1 x_1 + w_2 x_2 = 0$ for some $\boldsymbol{w} = (w_0, w_1, w_2)^\mathsf{T}$. The true decision boundary is characterized by $\boldsymbol{w}_{\text{true}} = (0, 1, 0)^\mathsf{T}$. We denote the learned decision boundary as $\hat{\boldsymbol{w}}$. Next, we compute an error metric between the learned and true decision boundaries as follows. We first normalize the learned decision boundary and define $\hat{\boldsymbol{w}}_{\text{norm}} = \frac{\hat{\boldsymbol{w}}}{\|\hat{\boldsymbol{w}}\|_2}$. We do not need to do this for $\boldsymbol{w}_{\text{true}}$ since it is already normalized. Then, we use the error metric

$$\text{err}(\hat{\boldsymbol{w}}, \boldsymbol{w}_{\text{true}}) = \min\left(\|\hat{\boldsymbol{w}}_{\text{norm}} - \boldsymbol{w}_{\text{true}}\|_2, \|\hat{\boldsymbol{w}}_{\text{norm}} + \boldsymbol{w}_{\text{true}}\|_2\right). \tag{53}$$

Including both terms in the minimum is necessary due to unidentifiability of the signs of $\hat{\boldsymbol{w}}$.

Figure 5A compares this error metric across the 5 model fits for each vEM method. It is clear that the modified vEM approach consistently outperforms the standard vEM approach in terms of more accurately estimating the decision boundary. In addition, the error for standard vEM has much higher variance, since the algorithm is prone to getting stuck in local maxima of the ELBO. Figures 5B-C display the learned versus true decision boundaries in the latent space for two selected runs. We select the run from each vEM approach which achieved the lowest decision boundary error metric, as denoted by solid dots in fig. 5A. We find that the model fit with standard vEM learns a decision boundary which noticeably deviates from the true boundary. On the other hand, the model with with modified vEM recovers the true boundary almost perfectly. This illustrates that our modified approach dramatically improves kernel hyperparameter estimation in practice and enables the gpSLDS to be much more interpretable in the latent space.

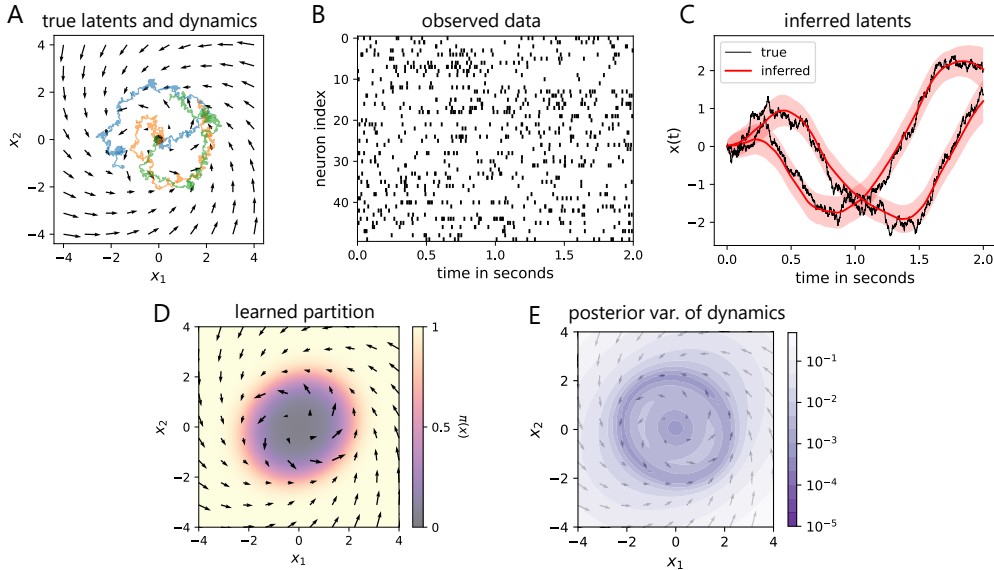

Figure 6: Additional synthetic data results on a 2D limit cycle from a gpSLDS fit with quadratic decision boundaries. **A.** True dynamics and true latent trajectories on 3 example trials used to generate the dataset. Dynamics are an unstable rotation and a stable rotation with fixed points at $(0,0)$, separated by $x_1^2 + x_2^2 = 4$. **B.** Poisson process observations for an example trial. **C.** gpSLDS inferred latent trajectory with $95\%$ posterior credible intervals for an example trial. **D.** The learned $\boldsymbol{\pi}(\boldsymbol{x})$ accurately recovers the true circular boundary between the two sets of linear dynamics. **E.** The gpSLDS learned posterior variance on dynamics. The posterior variance is low in regions heavily traversed by the true latent paths, and is high in regions with little to no data.

## D    Additional synthetic data results

To further demonstrate the expressivity of the gpSLDS over the rSLDS, we apply the gpSLDS to a synthetic dataset where the true decision boundary between linear regimes is nonlinear. The rSLDS can only model linear decision boundaries in order for its inference algorithm to remain tractable.

For this example, we generate a synthetic dataset consisting of an unstable linear system and a stable linear system separated by the decision boundary $x_1^2 + x_2^2 = 4$. Both of the linear systems have fixed points at $(0,0)$. The smooth combination of these linear systems results in a 2D limit cycle (fig. 6A). We simulate 30 trials of Poisson process observations from $D = 50$ neurons over $T = 2$ seconds (fig. 6B). We initialize the observation model parameters $\boldsymbol{C}$ and $\boldsymbol{d}$ using a Poisson LDS with data binned at 20ms. Then, we fit a gpSLDS with $J = 2$ regimes, and with $\boldsymbol{\pi}(\boldsymbol{x})$ modeled using the feature transformation

$$\boldsymbol{\phi}(\boldsymbol{x}) = \begin{bmatrix} 1 & \boldsymbol{x}_1^2 & \boldsymbol{x}_2^2 \end{bmatrix}^{\mathsf{T}}. \tag{54}$$

The results of this experiment are shown in fig. 6C-E. In fig. 6C we find that the gpSLDS successfully recovers the the true latent trajectory with accurate posterior credible intervals for an example trial. Furthermore, fig. 6D demonstrates that by using the quadratic feature transformation in eq. (54), the gpSLDS accurately learns the true flow field and true decision boundary $x_1^2 + x_2^2 = 4$. In addition, the values of $\boldsymbol{\pi}(\boldsymbol{x})$ smoothly transition between 0 and 1 near this boundary, highlighting the ability of our method to learn smooth dynamics if present. Lastly, in fig. 6E we plot the inferred posterior variance of our method. We find that the gpSLDS is more confident in regions of the latent space with more data (e.g. at the decision boundary), and less confident in regions of latent space with little to no data.

## E    Computing resources

We fit all of our models on a NVIDIA A100 GPU on an internal computing cluster. A breakdown of approximate compute times for the main experiments in this paper includes:

- Synthetic data results in Section 4.1: 1.5 hours per model fit, ~40 hours for the entire experiment.
- Real data results in Section 4.2: 1.5 hours per model fit, ~8 hours for the entire experiment.
- Real data results in Section 4.3: 1 hour per model fit, ~5 hours for the entire experiment.

We note that these estimates do not include the full set of experiments we performed while carrying out this project (such as preliminary or failed experiments).

