# OpenReview forum: "Modeling Latent Neural Dynamics with Gaussian Process Switching Linear Dynamical Systems"
_NeurIPS.cc/2024/Conference — NeurIPS 2024 poster_

### Official Review · Reviewer_3Mtv · 2024-06-13

**Soundness:** 4
**Presentation:** 4
**Contribution:** 3
**Rating:** 7
**Confidence:** 4

**Summary:**

This paper introduces a new approach to model the low-dimensional latent dynamics of a collection of neurons over time. Their approach balances the two desiderate of capturing complex nonlinear dynamics while remaining interpretable. Specifically, they introduce the Gaussian Process Switching Linear Dynamical System, which models the latent state as a Gaussian process with a specific novel kernel function, which interpolates between different linear dynamics akin to a switching linear dynamical system. This mitigates some of the pathologies of rSLDS and also provides uncertainty estimates for the dynamics. The authors validate their method both on synthetic and experimental data from neural recordings.

**Strengths:**

This paper tackles an important topic in neuroscience and provides interesting methodological advances continuing a line of research on inferring the latent dynamics of neuronal populations.
The paper is overall well-written and provides a concise and clear exposition.
For example, Figure 2G very clearly illustrates the problems that recurrent switching linear dynamical systems have.
The methodology seems sound to me and is motivated clearly in terms of the issues with rSLDS.
The experiments are well-executed and illustrative and importantly also use real data from two animal experiments tackling questions about behavior and decision-making. In addition, the authors provide code for their method and for two of the experiments.

**Weaknesses:**

The biggest weakness the paper currently has in my opinion is that it is hard to judge what the computational trade-offs are between the different methods. While the authors provide estimates for the overall compute they use in all experiments (see Appendix E), there is no comparison between different methods. I would have liked to see an analysis that compares GP-RBF, rSLDS and gpSLDS not only as a function of the number of trials or number of steps simulated forward, but also in terms of their predictive performance as a function of training time / budget. Otherwise, it is hard to judge whether these methods were compared fairly.

One aspect that would make the paper stronger, but is not strictly necessary in my opinion, is to add another baseline, which is different from the two approaches that were combined in gpSLDS, e.g. something like LFADS (Sussillo et al, 2016).

- Sussillo, D., Jozefowicz, R., Abbott, L., & Pandarinath, C. (2016). LFADS - Latent Factor Analysis via Dynamical Systems. In Advances in Neural Information Processing Systems (NeurIPS)

**Questions:**

- Figure 3: GP with RBF kernel seems to perform better with respect to forward simulation, why do you think that is?
- Could you comment on how expensive the different approaches are relative to each other in terms of training and prediction?

**Limitations:**

While the paper discusses limitations of their method throughout the paper, this is a bit sparse. I think the paper could be improved by a dedicated limitations paragraph in the discussion in Section 6.

---

> ### Author Rebuttal · Authors · 2024-08-07
>
> We thank the reviewer for taking the time to read our submission and for noting the strengths of our work! We are especially pleased that the reviewer praised our work's clear motivation, solid experimental results, and relevance to the neuroscience community.
>
> ## Weaknesses
> **Re. Runtime comparison**
>
> Thank you for this suggestion – we refer the reviewer to General Response A for new experimental results and a discussion about comparing runtimes and accuracy between the three methods. To further address your specific concerns:
>
> > predictive performance as a function of training time / budget
>
> Instead of comparing the predictive performance as a function of training time, we decided to compare the accuracy of the recovered latent states and dynamics and corresponding runtimes for a fixed number of vEM iterations sufficient for convergence. We did this for the following reasons:
>
> 1. Measuring how well the model recovers ground truth latent states and dynamics is the most direct way of evaluating the model’s performance on synthetic data, as we are primarily interested in how well the model captures the underlying latent variables.
>
> 2. For each model class, our aim is to fit the best possible model (in terms of ground truth recovery) using our available computing resources. Therefore, we first fit all models to convergence and then compared their runtimes and performance. We believe that this approach best reflects how these models are applied in practice. While investigating the learning dynamics as a function of fitting time is certainly an interesting question, it is somewhat orthogonal to the primary scientific goal of understanding latent neural dynamics.
>
> > hard to judge whether these methods were compared fairly
>
> Since we fit methods using different discretization steps to ensure the numerical stability in each model class, we report total runtime as well as runtime normalized by number of time bins. This allows us to fairly compare runtimes, which we expect to scale linearly with time bins in all 3 models.
>
> Finally, we note that training time depends on many factors specific to each individual implementation, and this should be taken into account when comparing runtimes. In General Response A & the attached PDF, we show that our implementation of the gpSLDS and other GP-SDE models – which leverages fast and efficient parallelization and auto differentiation in JAX – is more efficient per time bin than the most widely-used rSLDS implementation. We believe that this is a valuable contribution to the machine learning and neuroscience communities in itself, and we plan to release a codebase to the public along with the paper.
>
> **Re. LFADS**
>
> Thank you for the suggestion to compare our method with LFADS. We agree it would be interesting to explore this comparison in future work. On a high level, we expect LFADS to reconstruct neural data well, especially given sufficient data. However, the RNN dynamics of LFADS can be difficult to interpret after fitting and do not come with uncertainty estimates. Moreover, because LFADS is a deep learning model with many more parameters than the gpSLDS, we anticipate that it would struggle more to learn accurate dynamics in data-scarce settings, such as the hypothalamic dataset we analyze in our paper. In contrast, models with structured probabilistic priors, such as the gpSLDS, are better suited to correctly infer key dynamical motifs in these settings.
>
> ## Questions
> **Re. Comparison to GP-SDE w/ RBF kernel**
> > Figure 3: GP with RBF kernel seems to perform better with respect to forward simulation, why do you think that is?
>
> This is likely because the GP-SDE with RBF kernel is more expressive than the gpSLDS kernel, but it comes at the cost of being less interpretable. The RBF kernel is universal in the sense that it can approximate any smooth function with arbitrarily small error [Micchelli and Xu 2006]. However, because of its flexibility, key dynamical features such as fixed points are not readily available for downstream analysis. On the other hand, the gpSLDS finds the best (smoothly-interpolating) piecewise-linear approximation to a nonlinear system. This aids in interpretability by showing how nonlinear dynamics can be partitioned into linear regimes, each of which can be individually analyzed. Therefore, we expect that the gpSLDS finds dynamics which are less accurate but more representative of dynamical motifs of interest.
>
> **Re. Computational complexity and tradeoffs**
> > Could you comment on how expensive the different approaches are relative to each other in terms of training and prediction?
>
> We provide a discussion on the computational complexity of the gpSLDS in General Response A. All 3 methods use inference algorithms that scale linearly in sequence length. With respect to latent dimension, the gpSLDS scales exponentially (due to using quadrature) while the rSLDS scales cubically. However, the exponential scaling in the gpSLDS could be overcome by using Monte Carlo approximations instead of quadrature for large latent dimensionalities.
>
> In addition, all methods can predict dynamics at new locations efficiently once the models are fit. For the GP-SDE based models, predicting dynamics at a batch of $B$ new locations costs $O(KBM^2)$, where $M$ is the number of sparse inducing points. For the rSLDS, this operation costs $O(KB)$ by reading off the dynamics of the most likely discrete state. In practice, the main cost comes from fitting the model rather than from predicting dynamics. We will include a discussion about relative computational tradeoffs to our paper.
>
> **Re. Addressing limitations**
>
> Thanks for this suggestion – we have written a discussion on limitations and possible model extensions in General Response B, which we plan to add as its own paragraph in the paper.
>
> **Thank you again for your positive comments and insightful response!** If you have any further questions, we would be happy to answer them.
>
> – The gpSLDS authors

---

> > ### Comment · Reviewer_3Mtv · 2024-08-07
> >
> > Thank you for answering my questions on computational cost and adding an explicit limitations section. I believe the changes in the rebuttal strengthen the paper and have raised my confidence in my earlier assessment that this paper should be accepted. I've increased my confidence score to reflect this.

---

### Official Review · Reviewer_E7YR · 2024-07-11

**Soundness:** 2
**Presentation:** 3
**Contribution:** 3
**Rating:** 6
**Confidence:** 4

**Summary:**

This paper proposes a new model called Gaussian Process Switching Linear Dynamical System (gpSLDS). The model is more interpretable and infers more stable latent compared with the alternative rSLDS. Particularly, the weird oscillations of the latent can be avoided due to the newly proposed Smoothly Switching Linear (SSL) kernel. Extensive experimental results on one synthetic and two real-world datasets shows the benefit of gpSLDS compared with the one with the traditional RBF kernel and the rSLDS.

**Strengths:**

* This proposed method is detailed in math and intuitive with some instructive explanations.
* One synthetic and two real-world experiments are done, which is good.

**Weaknesses:**

* Some of the presentations can be improved.
* Lacks some complexity analysis or results. This might be an important factor to be considered for long sequences.

See questions.

**Questions:**

* Typo: Line 129, function's slope of
* Figure 1A, if different colors are different samples (should be clearly state if so), then it is meaningless to show these samples unless kernel parameters are told.
* Line 177, $\boldsymbol z_m \in \mathbb{R}^D$? And it seems like this $\boldsymbol z$ is different from the $z_j$.
* What does the conjugacy of the model mean in line 193?
* How about the effect of choosing $K$ and $J$. At least some explanations or guidance and the corresponding effects are necessary for them.
* How about the algorithm complexity. Computing GP is time consuming, which might hinder the computing efficiency for long sequences.

**Limitations:**

/

---

> ### Author Rebuttal · Authors · 2024-08-07
>
> We thank the reviewer for taking the time to read our submission and for providing helpful feedback! We are especially pleased that the reviewer found our methodology to be intuitive and supported by solid experimental results.
>
> ## Weaknesses
> **Re. Complexity analysis**
> > Lacks some complexity analysis or results. This might be an important factor to be considered for long sequences.
>
> We refer the reviewer to General Response A for an analysis and discussion of the computational complexity of the gpSLDS. We highlight that while typical GP inference methods scale as $O(T^3)$ where $T$ is the number of time steps, we overcome this by employing inducing points as in Titsias (2009) and Duncker et al. (2019) to reduce this complexity to $O(TM^2)$ where $M$ is the number of inducing points.  It is true, however, that the gpSLDS incurs larger computational costs in scaling with respect to latent dimension due to using quadrature to approximate kernel expectations. For many real world datasets, such as the ones we explore in our paper, the key dynamical features of interest can be captured in low latent dimensions. That said, there will be some applications which require large latent dimensionalities; for these cases, one possible workaround would be to use Monte Carlo methods instead of quadrature. We also highlight that in practice, our implementation of the gpSLDS leverages fast parallelization and auto-differentiation in JAX, which yields faster runtimes per time bin than the rSLDS (see General Response A & attached PDF).
>
> ## Questions
> **Re. Line 129**
>
> Thank you for catching this typo! We will be sure to fix this in our paper.
>
> **Re. Figure 1A**
>
> In Figure 1A, different colors do indeed represent different samples from the GP. We will make sure to clearly state this in the figure caption. We decided to show multiple samples for each GP kernel to provide intuition for what each distribution over functions looks like and how they build up to the SSL kernel.
>
> **Re. Line 177 notation**
>
> Thanks for catching this typo as well! Line 177 should say $\mathbf{z}_m \in \mathbb{R}^K$. Thank you also for pointing out the overloaded notation for $z$. We will keep the notation for the inducing inputs and will change the notation for the rSLDS discrete states to $s_j$.
>
> **Re. Model conjugacy**
>
> Here, model conjugacy refers to the fact that we can compute the distributions $q(\mathbf{u}_k)$ which maximizes the ELBO in closed-form. This is due to the conjugacy between the Gaussian prior on inducing points $p(\mathbf{u}_k \mid \Theta) = \mathcal{N}(\mathbf{u}\_k \mid 0, \mathbf{K}\_{zz})$ and its corresponding Gaussian variational posterior $q(\mathbf{u}\_k) = \mathcal{N}(\mathbf{u}\_k \mid \mathbf{m}\_u^{k}, \mathbf{S}_u^{k})$. We provide more detail about this update step in Appendix B.4, and present the closed-form updates for $\mathbf{m}\_u^{k}$ and $\mathbf{S}_u^{k}$ in Equations (40)-(41) of that section.
>
> **Re. Choosing $K$ and $J$**
>
> Thanks for this question. In cases where we do not know the true latent dimensionality or number of linear regimes, we can use standard model comparison metrics in the neural latent variable modeling literature to choose these hyperparameters. For example, we could compare models with different $K$ and $J$ based on forward simulation accuracy, which measures how well we can predict neural activity if we sample future latent states from the fitted model [Nassar et al. 2019; Nair et al. 2023]. Another option would be to use a validation technique called co-smoothing, which involves re-running the inference step of a fitted model on held-out trials after withholding some neurons, and then evaluating the expected log-likelihood on those withheld neurons [Macke et al. 2011; Wu et al. 2018; Keeley et al. 2020]. We will add a discussion on choosing $K$ and $J$ to our paper.
>
> **Thank you again for your positive comments and insightful response!** If you have any further questions, we would be happy to answer them.
>
> – The gpSLDS authors

---

> > ### Comment · Reviewer_E7YR · 2024-08-07
> >
> > Thanks for the response. I don't have further questions and I would like to keep my score.

---

### Official Review · Reviewer_9ds8 · 2024-07-11

**Soundness:** 4
**Presentation:** 3
**Contribution:** 3
**Rating:** 6
**Confidence:** 3

**Summary:**

The paper explores latent state inference and parameter learning within a switching stochastic dynamical system. In this context, the dynamics are represented by a stochastic differential equation, with the drift function modeled as a Gaussian process. Notably, the paper introduces a novel kernel for this Gaussian process—a mixture of linear functions that captures the system’s switching behavior. The inference process employs a variational inference framework. To validate its effectiveness, the paper conducts evaluations using both synthetic and real neuroscience data.

**Strengths:**

The proposed method offers a fresh perspective on inference for switching dynamical systems. It introduces a novel Gaussian process (GP) kernel that captures the switching behavior between linear functions. The experimental results are convincing and fair, and the paper is well-written and easy to follow.

Additionally, the authors suggest a minor modification to the variational inference (VI) algorithm to achieve faster updates.

**Weaknesses:**

My primary concern with this work lies in how the latent dynamics, denoted as $f$, are modeled as independent draws from the same Gaussian process (GP) prior. In my view, this implies that the posterior over latent components should be independent in each component. Consequently, learning becomes challenging when only single components of the system are observed. This limitation could pose a substantial problem, especially in classical tracking scenarios where only specific state components are observable.

Additionally, I perceive the contribution as somewhat marginal. Essentially, the paper introduces a new GP kernel, but beyond that, the impact seems limited.

Choice of Linear Kernel:

I wonder about the rationale behind using a linear kernel. Why not explore switching between nonlinear kernels?

Inference Algorithm Analysis:

Lastly, a detailed examination of the variational inference algorithm would provide valuable insights. Additionally, a synthetic example for comparison could help demonstrate the algorithm’s performance.

For instance, the authors could select a linear stochastic differential equation (SDE) and perform closed-form parameter estimation using methods like the Expectation Maximization (EM) algorithm. In this scenario, latent state inference could be achieved through RTS smoothing.

**Questions:**

- Could the authors provide further details on how the presented approach could incorporate a multi-dimensional correlated Gaussian process prior?

**Limitations:**

-

---

> ### Author Rebuttal · Authors · 2024-08-07
>
> We thank the reviewer for taking the time to read our submission and for providing insightful feedback. We especially appreciated that the reviewer found our submission to be clearly written and a "fresh perspective" on SLDS models!
>
> ## Weaknesses
> **Re. Prior independence assumption**\
> We thank the reviewer for this comment and agree it is important to consider how certain modeling choices might introduce estimation error or limit expressivity. Here, we provide a discussion on this point which we'll also include in the paper. We note that in our setting, we assume high-dimensional observations are driven by mixtures of latent components. Observing a lower-dimensional projection of the SDE where only single components are observed is not a setting we consider here, but would be an interesting extension.
>
> We model $f$ using independent GPs, following a body of previous work [Eleftheriadis et al. 2017; Duncker et al. 2019; Fan et al. 2023]. We note that although the prior assumes independence, **this does not imply that the true posterior over $f$ is independent across dimensions**, as the likelihood depends on the latent state $x$ which combines $f$ across dimensions. We approximate the true posterior using a variational approximation that factorizes over latent dimensions. This enables a tractable inference algorithm which propagates posterior uncertainty to the inference and prediction of latent dynamics. While we could add covariance terms to the variational approximation, this would introduce more parameters which may complicate inference and learning. Nonetheless, we agree it is important to carefully assess biases that the variational approximation may induce in the recovered estimates (e.g. Turner and Sahani 2011) and this will be a topic of future work.
>
> **Re. Impact of contribution**
> > the paper introduces a new GP kernel, but beyond that, the impact seems limited
>
> While we do introduce a new GP kernel, we believe its broader implications are significant for both the ML and neuroscience communities. We identified key limitations of the rSLDS and drew a nontrivial connection to GP-SDEs via the design of a novel kernel, retaining many of the advantages of the rSLDS while addressing its limitations. The gpSLDS extends a rich line of work on rSLDS and related models which have made a significant impact on interpretable data analysis in neuroscience [Taghia et al. 2018; Costa et al. 2019; Nair et al. 2023; Liu et al. 2023, Vinograd et al. 2024]. We have also developed a fast and efficient JAX codebase implementing the gpSLDS and other GP-SDE models. We will release our codebase to the public upon acceptance, providing practitioners with a valuable tool for modeling neural dynamics.
>
> **Re. Why linear kernel?**
> > I wonder about the rationale behind using a linear kernel. Why not explore switching between nonlinear kernels?
>
> We designed the gpSLDS to switch between linear kernels to impose interpretable structure on complex nonlinear dynamics so that they can be easily analyzed downstream. Typical analyses of nonlinear dynamics focus on linearized dynamics around fixed points [Duncker et al. 2019; Sussillo & Barak 2013], and often require second-stage analyses like fixed-point finding [Golub & Sussillo 2018; Smith et al. 2021]. In contrast, interpretable features are readily available in the gpSLDS due to its piecewise-linear structure. In neuroscience, dynamical motifs of linear systems are hypothesized to underlie various kinds of neural computations (e.g. line attractors for evidence integration, rotational dynamics for motor control) [Vyas et al. 2020]. Our goal is to extract these features from neural data in an interpretable way.
>
> While it is straightforward to extend our GP kernel to switch between nonlinear kernels instead, this added model flexibility would likely make it difficult to correctly learn regime boundaries. In principle, a sufficiently expressive nonlinear kernel may not need to switch at all. In our case, by allowing linear functions to switch as a learnable function of the latent state, we can capture complex nonlinearities in the dynamics while also adding structure to make parameter learning more feasible.
>
> **Re. Validating the variational inference alg.**
> > a synthetic example for comparison could help demonstrate the algorithm’s performance
>
> We note that our experiment in Appendix C shows the improved performance of our modified vEM algorithm over standard vEM on synthetic data. For this experiment, we used the same synthetic dataset as in Figure 2 of the main text (two linear systems separated by a vertical boundary).
> > The authors could select a linear SDE and perform closed-form parameter estimation...
>
> Thanks for this suggestion – while this would indeed allow us to compare our vEM algorithm to closed-form EM, our focus is on performing approximate posterior inference for nonlinear SDEs, for which closed-form updates are not available. Therefore, we performed a slightly different experiment in Appendix C: comparing our modified vEM algorithm to standard vEM on a dataset with simple, but nonlinear, dynamics. We chose this setting because the nonlinearity introduces complex dependencies between the dynamics and kernel hyperparameters that would not be present in the linear case.
>
> ## Questions
> > …incorporate a multi-dimensional correlated Gaussian process prior?
>
> As we discuss above, the prior independence assumption does not imply that the true posterior is independent across dimensions. In principle it is possible to incorporate correlation structure into the prior, but this would introduce $O(K^2)$ more model parameters that may be harder to learn. Since the true posterior is still correlated across dimensions even with an independent GP prior, we decided to stick with the simpler approach.
>
> **Thank you again for your positive comments and insightful response!** If you have any further questions, we would be happy to answer them.
>
> – The gpSLDS authors

---

> > ### Comment · Reviewer_9ds8 · 2024-08-11
> >
> > I have read the rebutal and thank the authors for their answers. I do not have any further questions and will keep my score.

---

### Official Review · Reviewer_rveQ · 2024-07-12

**Soundness:** 3
**Presentation:** 3
**Contribution:** 3
**Rating:** 7
**Confidence:** 3

**Summary:**

This paper introduces the Gaussian Process Switching Linear Dynamical System (gpSLDS), a novel approach for modeling latent neural dynamics. This model extends the Gaussian process stochastic differential equations framework by incorporating a new kernel function that supports smoothly interpolated locally linear dynamics. This innovation allows the gpSLDS to maintain the expressiveness needed for complex systems while enhancing interpretability, addressing the limitations of the rSLDS. The paper's contributions include the development of the gpSLDS model, introduction of a novel kernel function to balance expressiveness and interpretability, and a new learning algorithm that improves the accuracy of kernel hyperparameter estimation. The model's effectiveness is shown through applications to both synthetic and real neuroscience data, showing superior performance compared to existing methods like the rSLDS. This advancement provides a more robust framework for understanding neural computation and dynamics.

**Strengths:**

1. The gpSLDS introduces a novel kernel function within the Gaussian process framework that uniquely addresses the trade-off between model expressiveness and interpretability in the analysis of neural dynamics. This model contribution is original in the using of switching dynamics within Gaussian process, allowing for smoothly interpolated transitions between local linear regimes.

2.  The methodology presented in the paper is well-developed. The paper proposes a GP-SDE model with a well-conceived kernel that facilitates a locally linear interpretation of complex dynamic behaviors.

3. The paper is well-written with a clear structure that guides the reader through the problem motivation, model formulation, and experiments.

**Weaknesses:**

1. The proposed model effectively transforms a linear dynamical system into a Gaussian Process with a well-designed kernel. While this adaptation supports smoothly interpolated locally linear dynamics, it does compromise computational efficiency, shifting from linear time to cubic time cost.

**Questions:**

N/A

**Limitations:**

The authors should explore and discuss the limitations of the gpSLDS, including specific scenarios or conditions under which the model may underperform or fail.

---

> ### Author Rebuttal · Authors · 2024-08-07
>
> We thank the reviewer for taking the time to read our submission and for noting the strengths of our work! We are especially glad to see the reviewer’s comments on the originality and soundness of our method, as well as the clarity of our submission.
>
> A large part of the review centers around questions of computational efficiency and limitations of the gpSLDS. We refer the reviewer to the General Response, in which we address both of those themes. In addition, we include more specific responses to your individual review here.
>
> ## Weaknesses
> **Re. Modeling benefits vs. computational tradeoffs**
> > While this adaptation supports smoothly interpolated locally linear dynamics, it does compromise computational efficiency, shifting from linear time to cubic time cost.
>
> We provide a general discussion of the computational scaling of our algorithm in General Response A, showing that our algorithm still scales linearly in sequence length. However, it is true that it incurs larger computational costs in the scaling with respect to the latent state dimensionality.
>
> While the gpSLDS does introduce more computational complexity, it brings several key modeling advantages:
>
> 1. The gpSLDS can smoothly interpolate between locally linear dynamics. This maintains interpretability by allowing each component to be analyzed downstream using principles of linear systems, while achieving greater expressivity by being able to learn nonlinearities between these linear regimes. Crucially, this resolves problems commonly experienced in the rSLDS, such as artifactual oscillations of dynamics at regime boundaries (Fig. 2G).
>
> 2. Our GP-based approach allows us to infer approximate posterior distributions over dynamics at any point in the latent space, whereas the rSLDS often infers uninterpretable dynamics at regime boundaries and does not explicitly treat dynamics parameters as probabilistic quantities.
>
> We believe that these modeling advantages of the gpSLDS effectively address limitations in the rSLDS that hinder its interpretability in practice. In addition, our implementation of the gpSLDS and other GP-SDE based models leverages fast parallelization and automatic differentiation in JAX with GPU compatibility. This allows the gpSLDS to achieve more efficient runtimes on a per-timestep basis than the most widely-used rSLDS implementation (see General Response A & attached PDF), despite the additional complexity in GP inference. We will release a codebase with the paper if accepted, which we believe will provide a valuable and practical tool for practitioners.
>
> ## Limitations
> > The authors should explore and discuss the limitations of the gpSLDS, including specific scenarios or conditions under which the model may underperform or fail.
>
> We refer the reviewer to General Response B for a discussion on the limitations of the gpSLDS, as well as possible model extensions.
>
> **Thank you again for your positive comments and thoughtful response!** If you have any further questions, we would be happy to answer them.
>
> – The gpSLDS authors

---

> > ### Comment · Reviewer_rveQ · 2024-08-08
> >
> > Thank you for the response. I have no further questions and will keep my score.

---

### Author Rebuttal · Authors · 2024-08-07

We thank the reviewers for taking the time to read our submission and for providing thoughtful and insightful feedback! We were pleased that the reviewers unanimously supported our submission as a valuable contribution to the NeurIPS community, citing that it was **1) easy to follow, 2) clearly motivated, and 3) backed by convincing experimental results.**

Here, we address two main themes that were brought up in several of the reviews. We respond to individual reviewer concerns separately.

# (A) Computational complexity of gpSLDS
Reviewers rveQ, E7YR, and 3Mtv brought up questions about the computational complexity and efficiency of the gpSLDS.

Performing inference and hyperparameter learning in the gpSLDS relies on computing expectations of $f(\cdot)$ with respect to the variational marginals $q(x(t)) \sim \mathcal{N}(m(t), S(t))$ and the approximate posterior GP $q(f)$. GP-based methods are computationally expensive for two reasons: 1) they typically scale cubically in the number of input points to the GP, and 2) evaluating posterior expectations with respect to distributions over GP inputs involve computing expectations of nonlinear kernel functions, which are typically not available in closed form.

To overcome 1), we follow Titsias (2009) and Duncker et al. (2019) in using inducing points to perform inference of $q(f)$. This reduces the computational complexity of evaluating $f(\cdot)$ on a sequence of $T$ latent states from $O(KT^3)$ to $O(KTM^2)$ for $M$ inducing points (in our synthetic experiments we choose $M=16$ ($4 \times 4$ grid) for a latent dimensionality of $K=2$). For 2) we perform quadrature with $N$ nodes per latent dimension, so the total number of nodes needed to accurately approximate kernel expectations scales as $O(N^K)$ (for all experiments we use $N=6$). This represents the main computational bottleneck of the algorithm and can pose challenges for fitting the gpSLDS in settings with large latent dimensionality. In the real datasets we study in our paper, a small number of latent dimensions could sufficiently capture key dynamical features of interest, and we expect this to be the case in other applications as well. However, in cases that require larger latent states, it is possible to use Monte Carlo methods instead of quadrature to approximate kernel expectations. We will include a thorough discussion of computational complexity and possible model extensions to our paper.

Reviewer 3Mtv suggested doing a runtime comparison between gpSLDS, GP-SDE with RBF kernel, and rSLDS. **In the attached PDF**, we include additional experimental results comparing the runtime and accuracy for the three methods on the synthetic dataset from Fig. 2. For the gpSLDS and GP-SDE with RBF kernel, we choose a discretization step of 1ms, yielding $T = 2500$ time bins. For the rSLDS, we choose a discretization step of 20ms, yielding $T = 125$ time bins. We observed that using too large of a discretization step for the GP-SDE based methods and using too small of a discretization step for the rSLDS can in some cases lead to numerical instabilities, due to their respective continuous-time and discrete-time model formulations. Therefore, we chose these settings in order to fairly compare the three models. We ran each method with $K = 2$ for $100$ variational EM iterations, which allowed all three methods’ ELBOs to converge. To deal with the differing sequence lengths, we report both total runtime and runtime normalized by the sequence length. We report standard errors across 5 runs with different random initializations per model.

These results demonstrate a computational tradeoff between the GP-SDE based models and the rSLDS: while the GP-SDE based models require more time bins (via smaller discretization steps) to accurately approximate continuous-time dynamics, they also recover the true latent variables with a much higher degree of accuracy on this continuous-time point process dataset than the rSLDS. In addition, these results show that our own implementations of the gpSLDS and GP-SDE with RBF kernel are more efficient than the most widely-used rSLDS implementation on a per time-bin basis. That being said, we note that differences in runtime are impacted by differences in implementation, such as the discretization step size, GPU compatibility, and choice of optimizer, which vary across models. **We stress that the focus of our contribution is not to optimize for runtimes, but rather to find the best-fitting model possible in order to draw accurate and reliable scientific conclusions.**

# (B) Clearly addressing limitations
Reviewers rveQ and 3Mtv suggested more clearly presenting the limitations of the gpSLDS. We agree that this would be a valuable addition to the paper, and we will include a paragraph discussing the following points:

1. Quadrature and scalability: As mentioned above, because we use quadrature methods to compute kernel expectations that are not available in closed-form, it could be challenging to scale the gpSLDS to settings with large latent dimensionalities. One possible extension could be to explore using Monte Carlo methods to approximate these expectations in higher latent dimensions.

2. Inference algorithm alternatives: One area of potential improvement for the gpSLDS would be to incorporate more recent methods for latent state inference in GP-SDEs [Verma et al. (2024); Course and Nair (2024)]. In particular, Verma et al. (2024) proposed an algorithm for inferring $q(x)$ akin to gradient descent, which they show achieves faster and more stable convergence compared to the fixed-point iteration method we use in our paper [Archambeau et al. (2006)]. While Verma et al. did not originally consider inference over $q(f)$, their method could be directly plugged into the inference algorithm for the gpSLDS.

**Again, we thank all of the reviewers for their positive comments and constructive feedback!**

– The gpSLDS authors

---

### Decision · Program_Chairs · 2024-09-25

**Decision:**

Accept (poster)

**Comment:**

By taking a product of a linear kernel and a "partition" kernel, gpSLDS elegantly improves Duncker & Sahani's work on estimating locally linear latent dynamical systems. The paper is a key advance in the topic, and it is clearly written with detailed experiments.